# Nuclear cGAS restricts L1 retrotransposition by promoting TRIM41-mediated ORF2p ubiquitination and degradation

Zhengyi Zhen[1,2,3,5], Yu Chen [1,2,5], Haiyan Wang[1], Huanyin Tang[1], Haiping Zhang[1,2], Haipeng Liu[4], Ying Jiang [1,2] & Zhiyong Mao [1,2,3] ✉

Cyclic GMP–AMP synthase (cGAS), initially identified as a cytosolic DNA sensor, detects DNA fragments to trigger an innate immune response. Recently, accumulating evidence reveals the presence of cGAS within the nucleus. However, the biological functions of nuclear cGAS are not fully understood. Here, we demonstrate that nuclear cGAS represses LINE-1 (L1) retrotransposition to preserve genome integrity in human cells. Mechanistically, the E3 ligase TRIM41 interacts with and ubiquitinates ORF2p to influence its stability, and cGAS enhances the association of ORF2p with TRIM41, thereby promoting TRIM41-mediated ORF2p degradation and the suppression of L1 retrotransposition. In response to DNA damage, cGAS is phosphorylated at serine residues 120 and 305 by CHK2, which promotes cGAS-TRIM41 association, facilitating TRIM41-mediated ORF2p degradation. Moreover, we show that nuclear cGAS mediates the repression of L1 retrotransposition in senescent cells induced by DNA damage agents. We also identify several cancer-associated cGAS mutations that abolish the suppressive effect on L1 retrotransposition by disrupting the CHK2-cGAS-TRIM41-ORF2p regulatory axis. Together, these findings indicate that nuclear cGAS exhibits an inhibitory function in L1 retrotransposition which could provide avenues for future interventions in both aging and tumorigenesis.

Cytosolic double-stranded DNA originating from exogenous pathogens, endogenous mitochondria or the nucleus is a signal of danger to cells and therefore may activate the DNA sensor cGAS to catalyze the formation of 2,3-cGAMP, initiating the transduction of the STING-IRF3-IFN innate immunity signaling cascade[1]. Recently, a number of reports have indicated that cGAS is also a nuclear protein under specific biological conditions in different types of cells. For instance, DNA damage-induced translocation of cGAS to the nucleus suppresses DNA double-strand break (DSB) repair by homologous recombination (HR)

in both cancer cells and normal fibroblasts[2,3], and a recent study also demonstrated that nuclear soluble cGAS is necessary for detecting nuclear-replicating DNA viruses[4]. The presence of cGAS in the nucleus creates risks for the cell, as abundant chromatin DNA may aberrantly activate cGAS to initiate the innate immune response. Cells have evolved different mechanisms to prevent cGAS activation by chromatin DNA; these mechanisms include the nucleosome inhibition[5-8], BAF competitive DNA binding[9], mitotic kinase Aurora B- or CDK1-mediated phosphorylation of cGAS[10,11], and restriction of STING

[1]Shanghai Key Laboratory of Maternal Fetal Medicine, Clinical and Translational Research Center of Shanghai First Maternity and Infant Hospital, Frontier Science Center for Stem Cell Research, School of Life Sciences and Technology, Tongji University, 200092 Shanghai, China. [2]Shanghai Key Laboratory of Signaling and Disease Research, School of Life Sciences and Technology, Tongji University, Shanghai 200092, China. [3]Tsingtao Advanced Research Institute, Tongji University, Qingdao 266071, China. [4]Shanghai Key Laboratory of Tuberculosis, Shanghai Pulmonary Hospital, School of Medicine, Tongji University, Shanghai 200092, China. [5]These authors contributed equally: Zhengyi Zhen, Yu Chen. ✉e-mail: zhiyong_mao@tongji.edu.cn

activation[12]. However, why is cGAS not strictly limited to the cytosol to prevent potential danger? In fact, from an evolutionary perspective, cGAS predates the interferon-based immunity pathway, indicating that cGAS might retain critical roles in other biological processes. Indeed, recent work indicates that nuclear cGAS plays roles in stabilizing replication forks to maintain genome integrity[13] and safeguarding against mitotic chromosome end-to-end fusions by suppressing DSB repair to preserve genome stability[14].

LINE-1 (L1) is a non-long terminal repeat (non-LTR) retro-transposon element and accounts for nearly 17% of the human genome[15]. The human genome includes only ~80–100 copies of full-length L1, that is, the retrotransposition-competent human L1, and it can move and insert into novel genomic sites[16,17]. The expression and retrotransposition of L1 elements are often associated with a number of physiological events, such as aging and age-associated diseases, including cancer and neurodegenerative diseases[18]. Great efforts have been made to elucidate the regulatory mechanisms of L1 repression or activation at the transcriptional level[19–24], but less attention has been given to the posttranslational regulation of ORF1p and ORF2p, two proteins encoded by open reading frames in full-length L1 and are required for L1 retrotransposition[25–27]. The few studies performed to date on the posttranslational regulation of L1 have been mainly focused on ORF1p. For instance, ORF1p is phosphorylated at several serine and threonine amino acid residues, and its phosphorylation is important for L1 retrotransposition activity[28–30]. TEX19.1 promoted ORF1p ubiquitination and degradation by enhancing the activity of the E3 ligase UBR2 in mouse embryonic stem cells[31]. In addition, TREX1 altered the intracellular localization of ORF1p, stimulating its degradation[32]. However, the posttranslational regulation of ORF2p, which exhibits both reverse transcriptase (RT) activity and endonuclease (EN) activity, has not been extensively investigated.

L1 and DNA damage have a complex mutually influential relationship. ORF2p, which is crucial for L1 retrotransposition, causes DNA strand breaks and poses a threat to genome integrity through its endonuclease activity. Moreover, genotoxic stress can modulate the activity of L1 retrotransposition[33]. Notably, DNA damage also induces the nuclear translocation of cGAS[2], which has been reported to function as a DNA sensor of L1 cDNA to trigger the expression of the senescence-associated secretory phenotype (SASP) and thus promotes cellular senescence[34]. However, whether nuclear cGAS directly participates in the regulation of L1 retrotransposition and the potential mechanism of this action have not been characterized.

In this study, we find that nuclear cGAS represses L1 retrotransposition to preserve genome stability by binding to and promoting the interaction of the E3 ligase TRIM41 and L1 ORF2p, thereby facilitating TRIM41-mediated ubiquitination and degradation of ORF2p. Moreover, we find that upon the occurrence of DNA damage, the checkpoint kinase CHK2, a critical factor involved in the DNA damage response, interacts with and phosphorylates cGAS at S120 and S305 residues to promote the association of cGAS-TRIM41 and TRIM41-ORF2p. Importantly, we demonstrate that the phosphorylation levels of cGAS S120 and S305 residues are upregulated, and L1 retrotransposition is inhibited in DNA damage-induced senescent cells. Moreover, several cGAS mutants observed in cancer patients fail to inhibit L1 retrotransposition because they disrupt the CHK2-cGAS-TRIM41-ORF2p regulatory axis.

## Results

### cGAS inhibits L1 retrotransposition by reducing ORF2p protein levels
To examine whether cGAS is involved in regulating L1 retro-transposition, we used a well-characterized GFP-based reporter for quantifying the L1 retrotransposition efficiency (Supplementary Fig. 1a)[26,35,36] to examine the effect of cGAS on L1 retrotransposition in HeLa cells. We found that cGAS overexpression significantly suppressed L1 retrotransposition events in a dose-dependent manner (Fig. 1a, b). Knocking out cGAS in HeLa cells significantly increased the retrotransposition efficiency, by 2.2- and 2.5-fold (Fig. 1c, d). Using either siRNA or shRNA, we observed similar results in cells with cGAS depletion (Supplementary Fig. 1b, c). Restoration of cGAS in knockout cells abrogated the stimulatory effect on L1 retrotransposition (Fig. 1e, f). To further validate our findings, we analyzed the genomic L1 copy number by real-time qPCR with a pair of primers targeting the genomic ORF2 DNA content, as previously reported[37]. The data revealed that the relative L1 copy number was significantly increased in cGAS knockout HeLa cells, compared to that in the control cells (Fig. 1g). Moreover, an increase in the L1 copy number was also observed in the kidney and brain of Cgas knockout mice, relative to the wild-type mice (Fig. 1h, i). Collectively, these data indicated that L1 retrotransposition events occurred more frequently upon the depletion of cGAS. More importantly, data mining of a previously published source[38] revealed that cGAS mRNA levels were negatively correlated with L1 retrotransposition in lung cancer (P = 0.01) (Supplementary Fig. 1d), confirming that cGAS inhibits L1 retrotransposition. By over-expressing cGAS mutants that showed decreased DNA binding capacity (cGAS-C396A/C397A and cGAS-K407A/K411A) and cGAS mutants that were enzymatically dead (cGAS-E225A/D227A and cGAS-D319A), we demonstrated that cGAS inhibited L1 retrotransposition independent of its DNA-binding ability and canonical enzymatic activity (Supplementary Fig. 1e, f).

We then hypothesized that cGAS might negatively regulate the protein levels of two L1-encoded proteins, ORF1p and/or ORF2p, thereby restricting L1 retrotransposition. Indeed, we found that in cGAS-knockout or cGAS-knockdown cells, the ORF2p protein level was markedly elevated (Fig. 1j, k), but no change in the ORF1p protein level was observed (Supplementary Fig. 1g). Moreover, cGAS over-expression significantly reduced the protein level of ORF2p but not that of ORF1p (Fig. 1l and Supplementary Fig. 1h). In addition, SILAC experiments also indicated that ORF2p protein levels increased in cGAS knockout cells (Fig. 1m). Consequently, cGAS overexpression suppressed ORF2p-induced accumulation of γH2AX, as revealed by immunostaining experiments (Fig. 1n, o) and Western blot analysis (Supplementary Fig. 1i). A comet assay also indicated that cGAS over-expression attenuated the ORF2p-induced increase in genome instability (Fig. 1p, q). In contrast, cGAS deficiency resulted in elevated γH2AX levels and increased genome instability in cells overexpressing ORF2p (Fig. 1r–u and Supplementary Fig. 1j). Therefore, our data suggest that cGAS reduced ORF2p expression to inhibit L1 retrotransposition.

### cGAS promotes TRIM41-mediated ubiquitination and degradation of ORF2p
We then dissected the regulatory mechanisms through which cGAS negatively regulated ORF2p protein levels. In cells overexpressing cGAS, MG132, which inhibits proteasome-mediated protein degradation[39], attenuated the cGAS-mediated decline in ORF2p protein levels (Fig. 2a, b), indicating that cGAS is a potential regulator of ORF2p ubiquitination and degradation. Consistent with this, knocking out cGAS significantly increased the protein stability of ORF2p (Supplementary Fig. 2a). Then, reciprocal co-IP experiments revealed that cGAS and ORF2p interacted with each other in human cells (Supplementary Fig. 2b, c). An in vitro co-IP assay was also carried out to validate our finding (Supplementary Fig. 2d). We then sought to determine whether cGAS-mediated regulation of ORF2p protein stability is mediated by altering the ORF2p ubiquitination level. We found that cGAS overexpression markedly promoted the K48-linked ubiquitination level of ORF2p (Fig. 2c), and knocking out cGAS reduced the ORF2p ubiquitination level (Fig. 2d). These data strongly indicate that cGAS directly interacts with ORF2p to promote its K48-linked ubiquitination, thereby contributing to ORF2p degradation. However, it

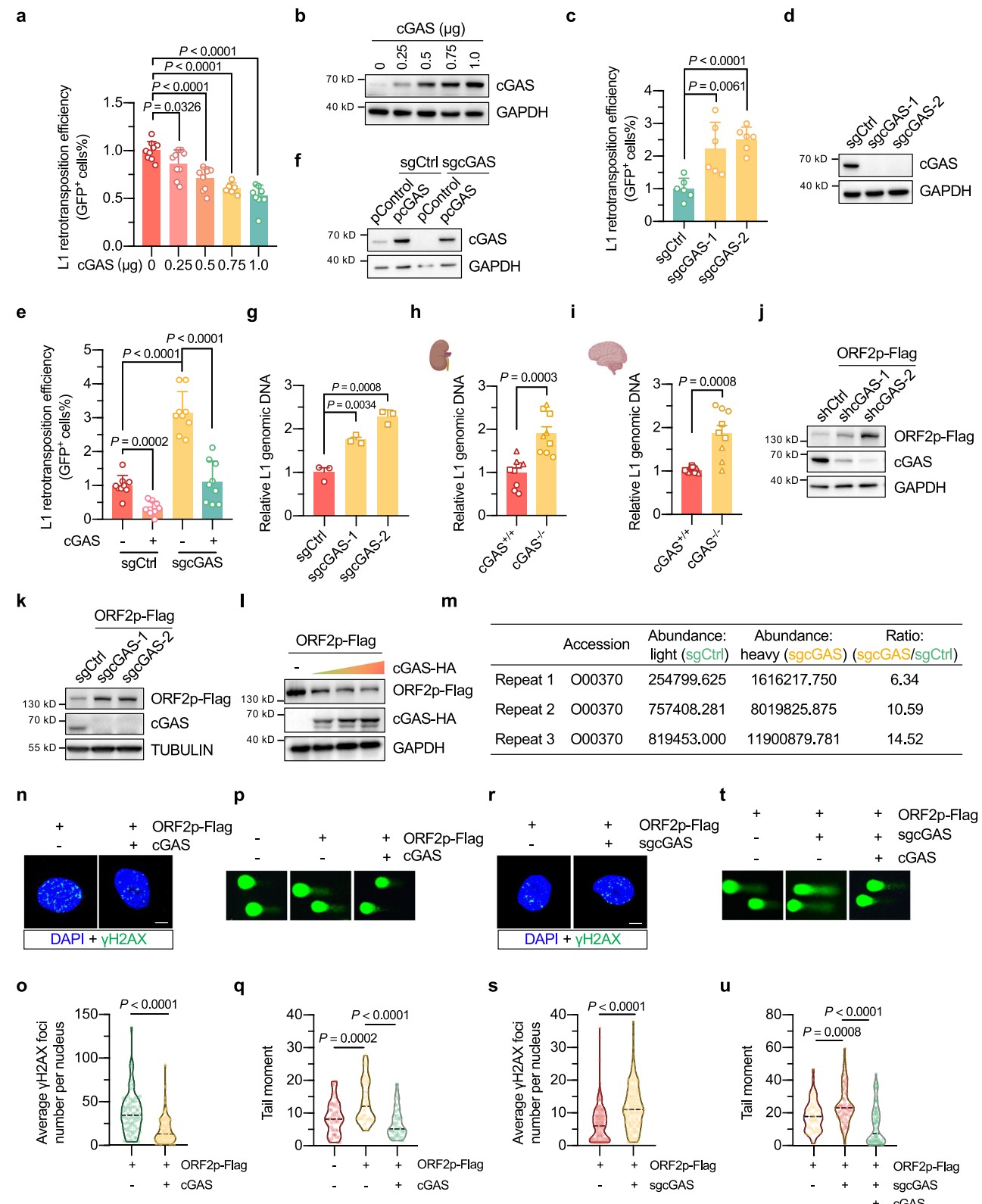

should be noted that although our subsequent focus is on studying the role of cGAS in regulating ORF2p protein levels, whether and how cGAS impacts L1 transcription necessitates further investigations in future work.

Since cGAS is not an enzyme that directly participates in the ubiquitination process, we hypothesized that cGAS might affect ORF2p ubiquitination by altering the interaction between ORF2p and its targeted E3 ligase. Unfortunately, the E3 ligase(s) that controls

ORF2p ubiquitination and degradation has not been identified. We therefore first sought to determine which E3 ligase(s) modifies ORF2p. Using lysates from cells with ectopic ORF2p-3×Flag expression, we performed a co-IP experiment with an antibody against Flag followed by mass spectrometry, which revealed 5 potential ORF2p-interacting E3 ligases: PIAS1, ZNF598, TRIM21, TRIM25 and TRIM41 (Fig. 2e). By analyzing the protein level of ORF2p in cells overexpressing each of these E3 ligases separately, we found that overexpressing TRIM41 but

**Fig. 1 | cGAS inhibits L1 retrotransposition by reducing ORF2p protein levels.**
**a**, **b** Effects of cGAS overexpression on L1 retrotransposition efficiency in HeLa cells. $n = 6$ independent experiments for the 0.75 μg group, and $n = 9$ for the others. **c**, **d** Analysis of L1 retrotransposition efficiency in cGAS-knockout HeLa cells. $n = 6$ independent experiments. **e**, **f** Restoring cGAS in cGAS-deficient cells abolished the stimulatory effect of knocking out cGAS on L1 retrotransposition. $n = 9$ independent experiments. **g** Analysis of endogenous L1 copy number in cGAS-knockout cells. $n = 3$ independent experiments. **h**, **i** Analysis of endogenous L1 copy number in the kidney and brain of *Cgas* knockout mice (9 replicates from three independent mice per group). **j**, **k** Analysis of the protein level of ORF2p in cGAS-depleted HeLa cells. **l** Analysis of ORF2p protein levels in HeLa cells transfected with cGAS-HA. **m** Relative ORF2p abundance between the heavy and light isotope labeling groups assayed by SILAC-combined mass spectrometry. **n**, **o** Immunofluorescence analysis of ORF2p-induced γH2AX foci with or without cGAS overexpression in HeLa cells.

The cells were transfected with a control vector or cGAS-HA vector together with an ORF2p-Flag vector. More than 50 cells were included for the analysis of each group (scale bar: 5 μm). **p**, **q** The effect of cGAS overexpression on genome stability in ORF2p-overexpressing HeLa cells. At least 50 cells per group were analyzed. **r**, **s** Immunofluorescence analysis of ORF2p-induced γH2AX foci in cGAS-knockout HeLa cells. More than 50 cells were included for the analysis of each group (scale bar: 5 μm). **t**, **u** The effect of knocking out cGAS on genome stability in ORF2p-overexpressing HeLa cells. At least 50 cells per group were analyzed. Data are presented as mean values ± s.d. for (**a**), (**c**), (**e**) and (**g**), and as mean values ± s.e.m. for (**h**) and (**i**). Student's $t$ test for (**a**), (**c**), (**g**), Brown-Forsythe and Welch ANOVA followed by Holm-Sidak's multiple comparisons for (**e**), and the Mann–Whitney $U$ test for (**h**), (**i**), (**o**), (**q**), (**s**) and (**u**). All statistical test used were two-sided. All experiments were repeated three times with similar results. Elements created with BioRender.com.

not the other 4 E3 ligases greatly reduced the ORF2p protein level (Fig. 2f), indicating that TRIM41 might be the E3 ligase regulating ORF2p ubiquitination and degradation. Reciprocal co-IP experiments demonstrated that ORF2p and TRIM41 interacted with each other in cells (Fig. 2g, h). Moreover, an in vitro co-IP experiment was also performed to verify our findings (Fig. 2i). We then created vectors to express different domains of the two aforementioned proteins (Supplementary Fig. 2e, f) and performed co-IP experiments to identify the interacting domains. We found that the EN domain of ORF2p was critical to ORF2p interaction with TRIM41 (Supplementary Fig. 2g), while the coiled-coil domain of TRIM41 was essential to the TRIM41 interaction with ORF2p (Supplementary Fig. 2h). Next, we examined the ubiquitination level of ORF2p in cells with TRIM41 overexpressed or knocked out. We found that TRIM41 overexpression increased ORF2p ubiquitination levels, while TRIM41 knockout inhibited ORF2p ubiquitination (Fig. 2j, k). These data indicate that TRIM41 is the E3 ligase regulating ORF2p ubiquitination and degradation.

Interestingly, a previous study showed that TRIM41 interacts with cGAS[40]. We therefore performed experiments to demonstrate that cGAS is involved in regulating TRIM41-mediated ORF2p ubiquitination and degradation. By performing co-IP and proximity ligation assay (PLA), we found that cGAS overexpression promoted the TRIM41-ORF2p interaction (Supplementary Fig. 2i–k). Moreover, knocking down or out cGAS expression attenuated the TRIM41-ORF2p interaction (Supplementary Fig. 2l–n). Importantly, knocking out TRIM41 abrogated the cGAS-mediated stimulatory effect on K48-linked ubiquitination and the cGAS-mediated inhibitory effect on ORF2p protein levels and retrotransposition efficiency (Fig. 2l–n), indicating that cGAS promotes ORF2p degradation by enhancing the TRIM41-ORF2p interaction and the subsequent ubiquitination–degradation process. In addition, we ruled out the possibility that cGAS might regulate L1 retrotransposition by inhibiting the formation of the ORF1p trimer (Supplementary Fig. 2o) and disrupting the ORF1p-ORF2p interaction (Supplementary Fig. 2p).

## CHK2 phosphorylates nuclear cGAS$^{S120/S305}$ to potentiate its repressive effect on L1 retrotransposition upon DNA damage

Our previous work demonstrated that cGAS is translocated to the nucleus upon DNA damage, thereby inhibiting HR repair[2]. We then hypothesized that the regulation of ORF2p protein levels by cGAS occurs in the nucleus and is dependent on the DNA damage response. Indeed, the cGAS Y215E mutant, which could not be translocated to the nucleus after DNA damage, showed no inhibitory effect on L1 retrotransposition (Fig. 3a). In contrast, the cGAS Y215F mutant, which resided in the nucleus, showed a repressive effect (Fig. 3a and Supplementary Fig. 3a). Subcellular fractionation followed by co-IP experiments demonstrated that the cGAS-ORF2p interaction occurred in the nucleus but not the cytosol (Fig. 3b). Further PLA assay also showed that the interaction between nuclear cGAS and ORF2p significantly increased upon the occurrence of DNA damage

(Supplementary Fig. 3b). These data indicate that nuclear cGAS promotes TRIM41-mediated ORF2p ubiquitination and degradation.

Then, we examined whether the regulation of ORF2p protein levels by cGAS depended on the DNA damage response by treating cells with different DNA damage sensor inhibitors, followed by Western blot analysis. We found that both ATM and CHK2 inhibitors but not ATR or CHK1 inhibitors abrogated the inhibitory effect of cGAS on ORF2p protein levels in a dose-dependent manner (Fig. 3c and Supplementary Fig. 3c, d). Given that ATM is the kinase that activates CHK2 kinase activity, the data suggest that either CHK2 or both kinases regulate ORF2p protein stability by phosphorylating cGAS. To determine which kinase(s) are involved in the regulation, we examined how an ATM inhibitor affected the ORF2p protein level in cGAS-overexpressing cells with CHK2 depletion. We found that depleting CHK2 abolished the effect of the ATM inhibitor on cGAS-mediated changes in ORF2p protein levels (Fig. 3d). Furthermore, co-IP experiment demonstrated that cGAS interacted with CHK2 (Fig. 3e), and DNA damage may promote the cGAS-CHK2 association (Fig. 3f), while the mutation D347A, which inactivates CHK2 kinase activity[41], drastically impaired the cGAS-CHK2 interaction (Fig. 3g). Consistently, the PLA experiment also indicated that the cGAS-CHK2 association exhibited a stress-dependent enhancement (Supplementary Fig. 3e). Moreover, a fluorescence-based Western blot was carried out, and the data further confirmed an enhanced association between cGAS and CHK2, but not the D347A mutant, in response to DNA damage (Supplementary Fig. 3f). In addition, although an interaction was observed between cGAS and ATM, the association was not further enhanced in response to etoposide treatment (Supplementary Fig. 3g). These data indicate that CHK2 but not ATM directly participates in the regulation of cGAS-mediated ORF2p degradation.

Next, we sought to identify the potential amino acid residue(s) in cGAS that is modified by CHK2. Since CHK2-mediated phosphorylation occurs mainly in the RXXS/RXXT motif[42], three amino acid residues, S120, T130 and S305, in cGAS are potentially phosphorylated by the kinase CHK2. We separately introduced mutations into these three potential phosphorylation sites and analyzed whether any of the mutations abolished the cGAS-mediated inhibitory effect on L1 retrotransposition. We found that the S120A and S305A mutations but not the T130A mutation in cGAS partially abrogated its suppressive effect and that these two mutations in combination further reduced the inhibitory effect compared to that of either mutation alone (Fig. 3h), suggesting that CHK2 phosphorylated cGAS on S120 and S305. Performed with an antibody we generated to recognize the phosphorylated S120 amino acid residue (Supplementary Fig. 3h) and a commercially available antibody against the phosphorylated S305 amino acid residue (Supplementary Fig. 3i), in vitro phosphorylation assays confirmed the CHK2-mediated phosphorylation of the two aforementioned amino acid residues in cGAS (Fig. 3i). Overexpressing CHK2 promoted the phosphorylation of cGAS at the two residues (Fig. 3j), and depleting CHK2 abrogated the DNA damage-dependent

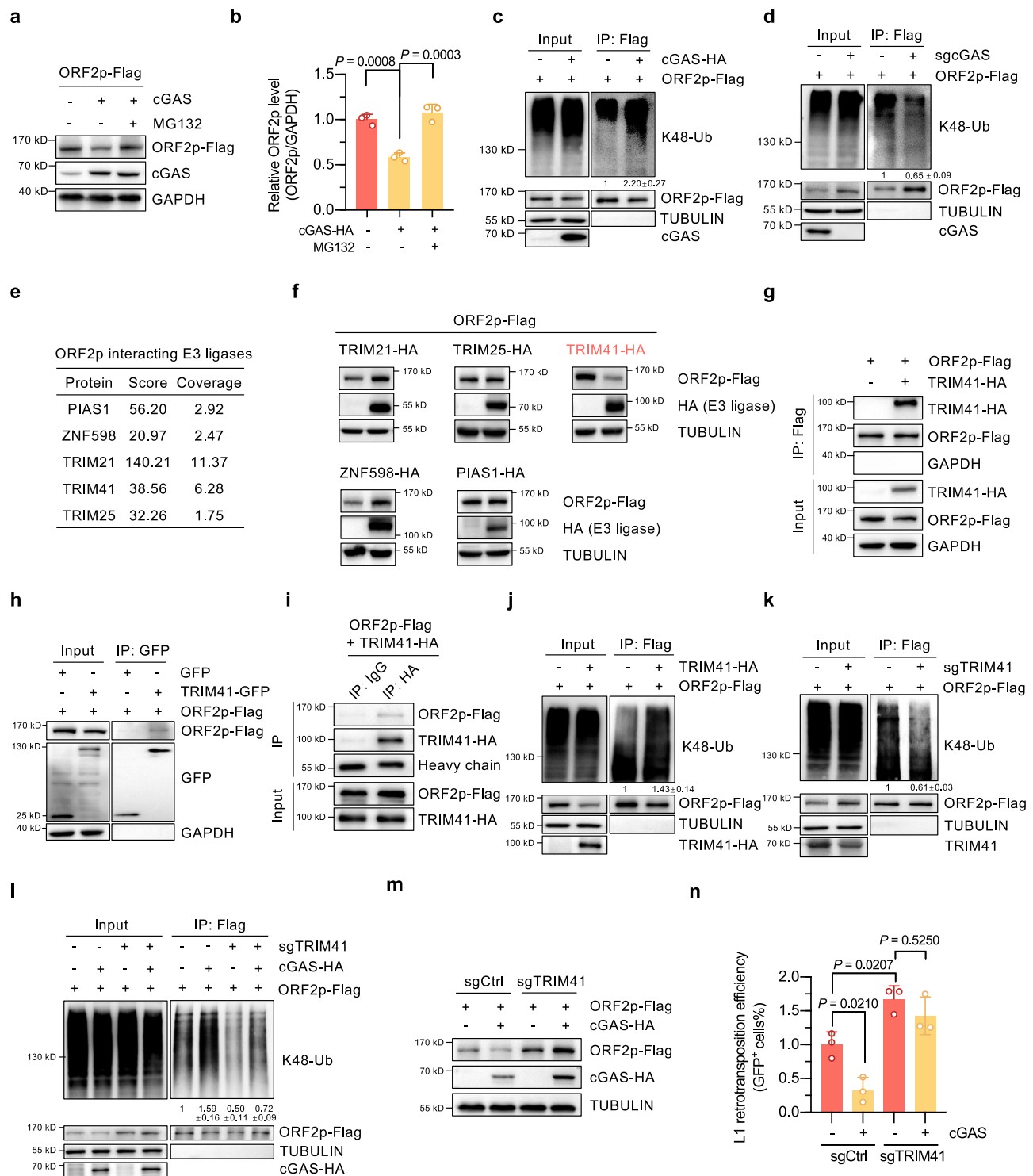

increase in cGAS phosphorylation levels (Fig. 3k and Supplementary Fig. 3j).

Since cGAS interacted with both TRIM41 and ORF2p, we set out to determine whether the aforementioned cGAS mutations would affect the cGAS-TRIM41 interaction or cGAS-ORF2p interaction. Co-IP and PLA assays revealed that these two mutations diminished the cGAS-TRIM41 interaction (Fig. 3l and Supplementary Fig. 4a, b) but not the cGAS-ORF2p interaction (Supplementary Fig. 4c). Consistently, both of the two phosphomimetic mutations of cGAS, S305E and S120E, enhanced the cGAS-TRIM41 association (Supplementary Fig. 4d, e). As a consequence, we observed that the two cGAS mutations abolished the cGAS-mediated stimulatory effect by interfering with the TRIM41-

ORF2p interaction (Fig. 3m and Supplementary Fig. 4f, g) and that the overexpressed cGAS S120A/S305A double mutants failed to exert an inhibitory effect on ORF2p protein level (Fig. 3n). In addition, we provided evidence that treating cells with BML-277, a CHK2 inhibitor, directly disrupted the enhanced association of cGAS with TRIM41 (Supplementary Fig. 5a–c) after DNA was damaged, and the stress-dependent increase of interaction between the two aforementioned proteins did not arise from increased nuclear localization of TRIM41 (Supplementary Fig. 5d). We also ruled out the possibility that CHK2-mediated enhancement in cGAS-TRIM41 interaction was caused by influencing the nuclear localization of cGAS (Supplementary Fig. 5e). Moreover, our data showed that the association of TRIM41 and ORF2p

**Fig. 2 | cGAS promotes TRIM41-mediated ubiquitination and degradation of ORF2p. a, b** cGAS promotes ORF2p degradation via the proteasome pathway. Six hours post-transfection with the indicated plasmids, cells were treated with 10 μM MG132 for 12 h, followed by Western blot analysis. **c** Analysis of ORF2p ubiquitination levels in HeLa cells overexpressing cGAS. Six hours post-transfection of a control vector or cGAS expression vector together with an ORF2p-Flag vector, cells were treated with 10 μM MG132 for 18 h and then lysed for use in co-IP and Western blot experiments. **d** The effect of knocking out cGAS on the ORF2p ubiquitination level. cGAS WT and knockout cells were transfected with the ORF2p-Flag vector. Then, 6 h post-transfection, the cells were treated with 10 μM MG132 and lysed for use in co-IP and Western blot experiments 24 h post-transfection. **e** Identification of ORF2p-interacting E3 ligases via mass spectrometry analysis. **f** Analysis of ORF2p protein levels in cells overexpressing the potential E3 ligases. Vectors encoding HA-tagged PIAS1, TRIM21, TRIM25, TRIM41 or ZNF598 and the

ORF2p-Flag expression plasmids were transfected into HEK293T cells, followed by protein extraction 24 h post-transfection. **g, h** Reciprocal co-IP analysis showed the interaction between TRIM41 and ORF2p in HEK293T cells. **i** Analysis of the interaction between TRIM41 and ORF2p in vitro with purified HA-tagged TRIM41 and Flag-tagged ORF2p proteins. **j, k** The effect of TRIM41 overexpression and knockout on ORF2p ubiquitination levels in HeLa cells. **l** The effect of cGAS overexpression on ORF2p ubiquitination levels in TRIM41-knockout cells. **m** Western blot analysis of ORF2p protein levels in cells transfected with sgRNAs against TRIM41 and/or vectors encoding cGAS. **n** Analysis of L1 retrotransposition efficiency in cells transfected with sgRNAs against TRIM41 and/or vectors encoding cGAS. $n = 3$ independent experiments. Data are presented as mean values ± s.d. One way ANOVA followed by Tukey's multiple comparisons for (**b**) and (**n**). All the Inputs and IPs were from the same experiments. Experiments were repeated three times independently.

---

increased when DNA was damaged, and the enhancement could also be disrupted by treating cells with BML-277 (Supplementary Fig. 5f, g). PLA experiments were also carried out to further validate that CHK2 inhibitor disrupted the enhanced interaction between TRIM41 and ORF2p upon the occurrence of DNA damage (Supplementary Fig. 5h). However, BML-277 treatment did not impact cGAS-ORF2p interaction (Supplementary Fig. 5i). Interestingly, our previous report indicated that cGAS interacted with PAR to suppress the formation of the PARP1-Timeless complex[2]; therefore, we hypothesized that cGAS might interact with ORF2p in a PAR-dependent manner. Indeed, the stress-dependent enhancement of the cGAS-ORF2p interaction was abolished by the PARP inhibitor olaparib (Supplementary Fig. 5j, k). Collectively, our results demonstrate that in response to DNA damage, CHK2 phosphorylates cGAS at the S120 and S305 residues to enhance its interaction with TRIM41, which promotes TRIM41-ORF2p interaction and eventually results in an increase in the K48-linked ubiquitination and degradation of ORF2p and the repression of L1 retrotransposition. However, it is of note that we observed the cGAS-mediated repressive effect of on L1 retrotransposition even in the absence of external DNA damage induction. We hypothesized that this impact may be attributed to a subset of cells experiencing endogenous DNA damage resulting from factors such as cellular proliferation or reactive oxygen species.

## cGAS is critical for senescence-associated repression in L1 retrotransposition

During cellular senescence, L1 becomes transcriptionally derepressed[34,43]; however, L1 retrotransposition is, surprisingly, repressed[44]. We hypothesized that cGAS might play a role in the retrotransposition repression in senescent cells, which are characterized by a destabilized genome and an activated DNA damage response signaling cascade[45,46]. We therefore induced HeLa cells to enter a state of senescence using the DNA damage inducer etoposide, and analyzed the efficiency of L1 retrotransposition in control and cGAS-deficient cells. We found that knocking out cGAS attenuated the decline in L1 retrotransposition in senescent HeLa cells (Fig. 4a, Supplementary Fig. 6a–e). Moreover, in senescent HeLa cells overexpressing cGAS, L1 retrotransposition proceeded at a negligible rate (Fig. 4b). We also performed real-time qPCR analysis to assess the genomic L1 copy number in senescent cells. Our results demonstrated that control wildtype cells exhibited an increase in genomic L1 DNA content compared to senescent cells. However, upon depletion of cGAS in senescent cells, we observed a small, but statistically significant increase in the genomic L1 DNA content, compared to wildtype senescent cells (Supplementary Fig. 6f, g). In senescent HeLa cells, an increase in CHK2 phosphorylation levels, cGAS S120 phosphorylation and cGAS S305 phosphorylation levels was observed (Fig. 4c). Further subcellular fractionation followed by Western blot analysis revealed that the observed increase in cGAS phosphorylation occurred in the nucleus, not the cytoplasm, in senescent HeLa cells (Fig. 4d). The upregulation in nuclear cGAS phosphorylation was confirmed in senescent IMR90-

hTERT (Fig. 4e) and HCA2-hTERT cells (Fig. 4f). Hence, our results suggest that activated CHK2 induces an increase in the phosphorylation level of nuclear cGAS at the S120 and S305 sites to inhibit L1 retrotransposition in senescent cells.

## Seven cancer-associated cGAS mutations abolish the repressive effect of cGAS on L1 retrotransposition

In addition, we systematically analyzed the cancer-associated cGAS mutations through data mining (Fig. 5a). We cloned 37 cGAS mutants and examined the effect of overexpressing cGAS mutants on L1 retrotransposition and HR repair. We found that none of the mutations significantly altered the suppressive effects of cGAS on HR (Supplementary Fig. 7a, b). However, seven of these cGAS mutations abolished the repressive effect on L1 retrotransposition (Fig. 5b), but they did not suppress their canonical immune regulatory function (Supplementary Fig. 7c). Importantly, we found that the seven mutants that had lost the ability to inhibit L1 retrotransposition also failed to reduce ORF2p protein levels (Fig. 5c). In comparison to WT-cGAS, three mutations (P486L, L377P and S345L) led to a decreased association between cGAS and CHK2 (Fig. 5d and Supplementary Fig. 7d). As a consequence, DNA damage-induced enhancement of the cGAS-TRIM41 interaction was abolished by each of the three aforementioned mutations (Fig. 5e and Supplementary Fig. 7e). Surprisingly, although the interaction between cGAS D408N or E383K and CHK2 did not change, we observed a reduction in the interaction between the two cGAS mutants and TRIM41 in response to DNA damage (Fig. 5e and Supplementary Fig. 7e). We propose that these two mutations might cause structural changes that disrupt the interaction between phosphorylated cGAS and TRIM41. Moreover, E216D, F433L and P486L, in contrast to the four other mutations (S345L, L377P, E383K and D408N), attenuated the cGAS-ORF2p interaction (Fig. 5f and Supplementary Fig. 7f, g). Therefore, all seven mutants failed to promote the TRIM41-ORF2p interaction (Fig. 5g and Supplementary Fig. 7h, i).

## Discussion

cGAS has recently been shown to interact with nucleosomes, DNA replication forks, DNA damage sites, centromeres and L1 sequences in the nucleus[2,3,13,47,48]; therefore, discovering the cGAS nuclear functions has become an intriguing topic in the field of cGAS biology. Most of the recently revealed nuclear functions of cGAS have led to positive outcomes for cells and organisms, as cGAS may decelerate the progression of the replication fork to stabilize the genome and initiate the innate immune response independent of its DNA-sensing function[13,49]. However, our previous study revealed that the DNA damage-induced translocalization of cGAS impaired DNA repair by HR[2]. Whether the DNA damage-triggered translocalization of cGAS plays a beneficial role was unclear. Our novel finding showing that nuclear cGAS repressed L1 retrotransposition explains the reason that cGAS nuclear translocation is necessary (Fig. 6). After DNA is damaged, chromatin is increasingly relaxed, which facilitates the completion of DNA repair[50]; however,

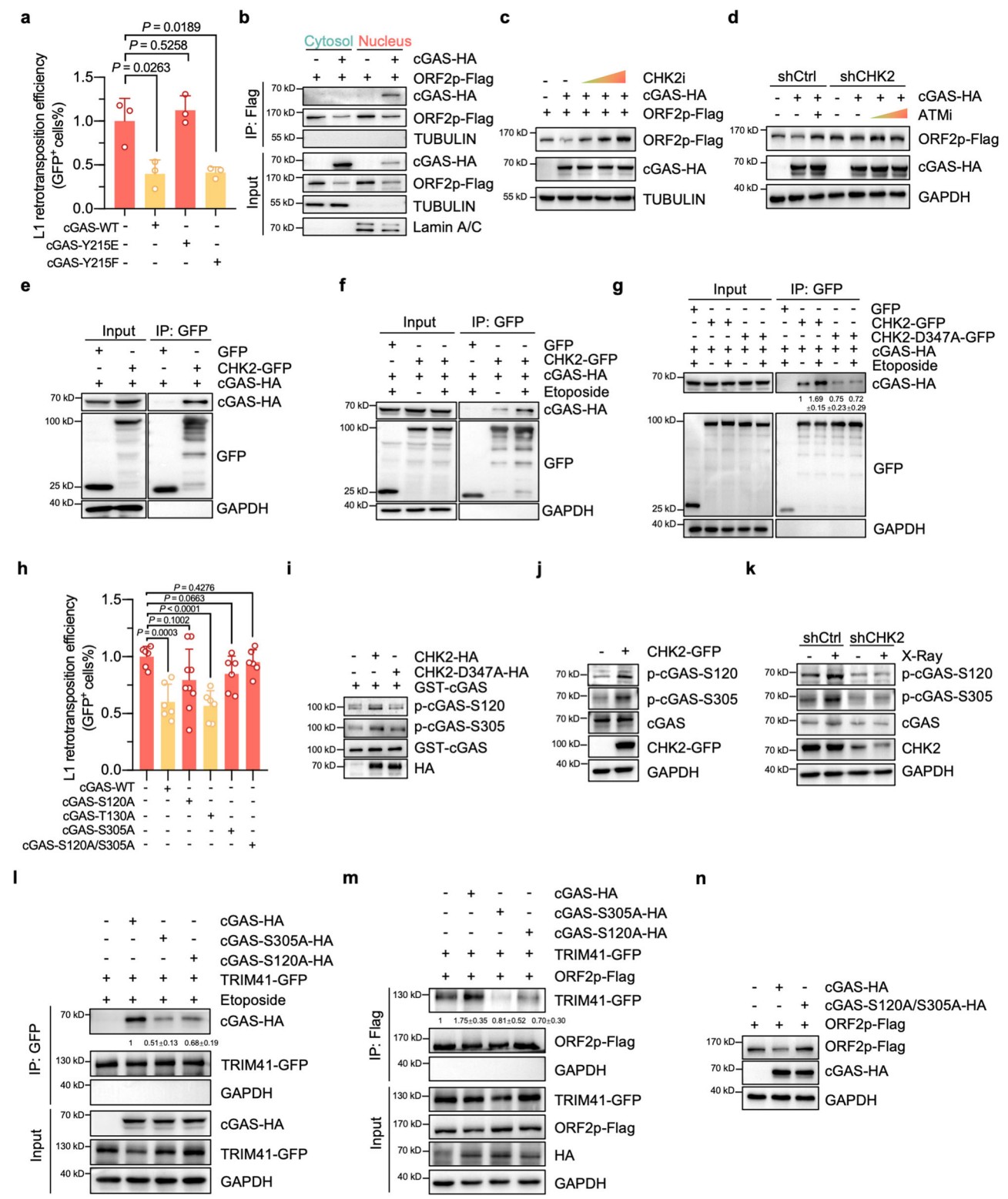

because the DNA around damage sites is more accessible to ORF2p, which may create DNA damage by cutting DNA strands. We propose that DNA damage response-associated regulation of cGAS plays a protective role in stopping ORF2p from digesting DNA and the subsequent L1 retrotransposition near DNA damage sites. Notably, our previous report indicated that cGAS directly interacted with the PAR chain and that this interaction impeded the formation of the PARP1-Timeless complex, thereby suppressing HR repair[2]. In this work, we

demonstrated that inhibiting PARP enzyme activity via olaparib treatment abolished the DNA damage-induced enhancement in cGAS and ORF2p interaction, indicating that the cGAS-PAR interaction may act as a double-edged sword by both suppressing HR to destabilize the genome and repressing L1 retrotransposition to maintain genome integrity.

Our current findings revealed a noncanonical function of cGAS, which it could inhibit L1 retrotransposition to stabilize the genome.

**Fig. 3 | CHK2 involves in nuclear cGAS-mediated ORF2p degradation and phosphorylates cGAS^S120/S305 to potentiate its repressive effect on L1 retro-transposition upon DNA damage. a** Analysis of L1 retrotransposition efficiency in HeLa cells expressing cGAS WT or mutants. *n* = 3 independent experiments. **b** cGAS interacts with ORF2p in the nucleus. **c** The effect of a CHK2 inhibitor on cGAS-mediated downregulation at the ORF2p protein level. Cells were treated with the CHK2 inhibitor 6 h post-transfection. **d** The effect of an ATM inhibitor on cGAS-mediated repression of ORF2p protein levels in CHK2-knockdown HeLa cells. Cells were treated with ATM inhibitor 6 h post-transfection. **e** Co-IP analysis of the interaction between cGAS and CHK2 in HEK293T cells. **f** Co-IP analysis showing the interaction between cGAS and CHK2 in HEK293T cells when the DNA was damaged. Cells were treated with 100 μg/mL etoposide for 4 h. **g** Comparison of the inter-action between cGAS and CHK2-WT or CHK2-D347A. **h** Analysis of L1 retro-transposition efficiency in cells overexpressing WT cGAS and the S120A, T130A and

S305A mutants. *n* = 9 independent experiments for the S120A group, and *n* = 6 independent experiments for the other groups. **i** In vitro analysis of CHK2-mediated phosphorylation of cGAS at the S120 or S305 residue. **j** The effect of CHK2 over-expression on cGAS phosphorylation levels at the S120 or S305 residue. **k** The effect of CHK2 depletion on cGAS phosphorylation levels at the S120 or S305 residues after DNA was damaged. Cells were treated with 10 Gy X-ray and lysed 2 h post irradiation. **l** Comparison of the interaction between TRIM41 and cGAS or its mutants after DNA was damaged. Cells were treated with 100 μg/mL etoposide for 4 h. **m** The effect of overexpressing cGAS or its mutants on the TRIM41-ORF2p interaction in HEK293T cells. **n** The effect of overexpressing cGAS or its mutants on ORF2p protein levels. Data are presented as mean values ± s.d. Student's *t* test was performed for (**a**) and (**h**). All the Inputs and IPs were from the same experiments. All statistical test used were two-sided. Experiments were repeated three times independently with similar results.

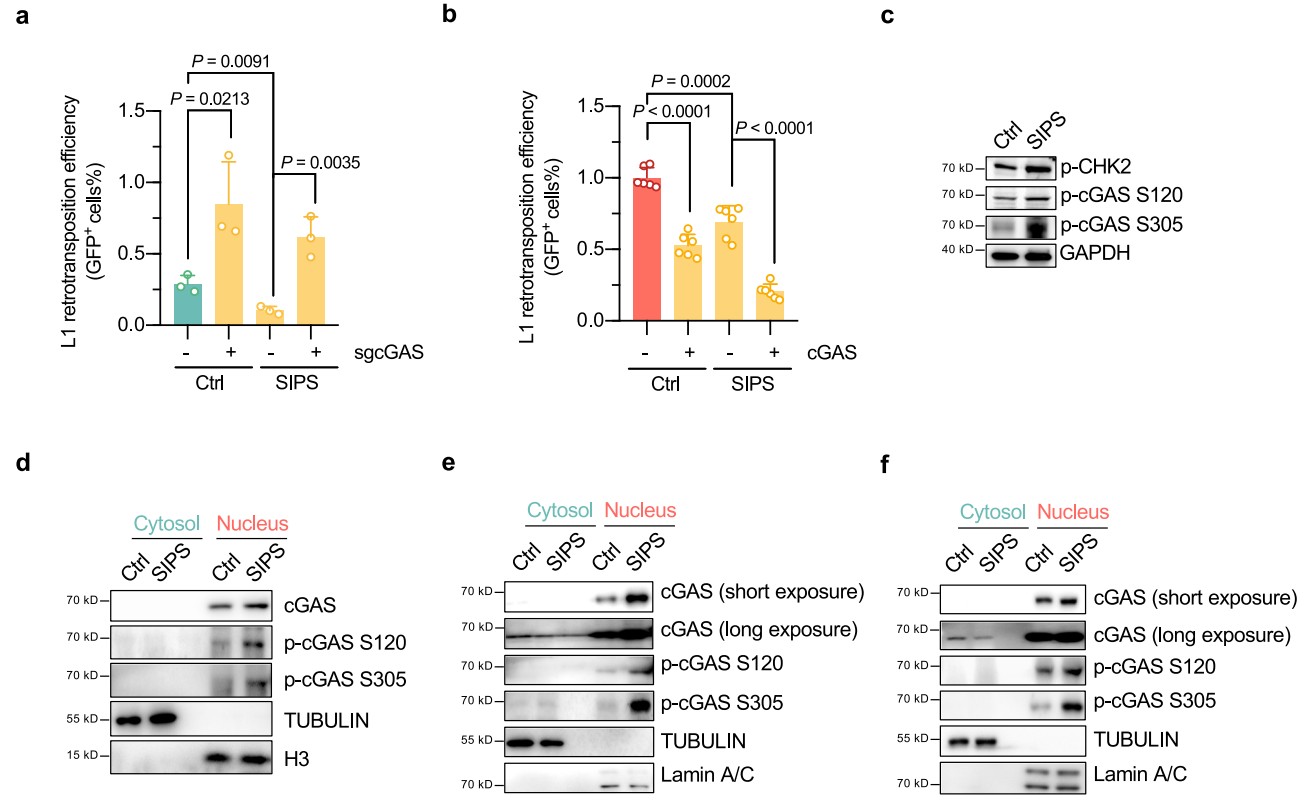

**Fig. 4 | cGAS is critical for senescence-associated repression in L1 retro-transposition. a** Analysis of the L1 retrotransposition efficiency in control or cGAS-knockout stress-induced premature senescent (SIPS) HeLa cells. WT or cGAS-knockout cells transfected with the pEGFP-LRE3 reporter were treated with 10 μg/mL etoposide for 20 min on day 2 post-transfection, and harvested for FACS ana-lysis on day 9. *n* = 3 independent experiments. **b** Analysis of L1 retrotransposition efficiency in control or cGAS-overexpressed SIPS cells. *n* = 6 independent experi-ments. **c** Analysis of CHK2 and cGAS phosphorylation levels in senescent HeLa cells.

**d** Subcellular localization of phosphorylated cGAS in senescent HeLa cells. **e** Analysis of the subcellular localization of phosphorylated cGAS in X-ray induced senescent IMR90-hTERT cells. **f** Analysis of the subcellular localization of phos-phorylated cGAS in X-ray induced senescent HCA2-hTERT cells. For (**e**) and (**f**), cells were irradiated by 15 Gy X-ray and lysed for protein extraction and Western blot analysis on day 9 post irradiation. Data are presented as mean values ± s.d. Stu-dent's t test was performed for (**a**) and (**b**). All statistical test used were two-sided. All experiments were repeated three times.

Furthermore, some cancer-associated cGAS mutations abolished the effect of cGAS on L1 repression through disrupting different steps of the CHK2-cGAS-TRIM41-ORF2p regulatory axis, suggesting that cGAS may act as a tumor suppressor. Consistent with this, elevated L1 expression has been reported in some types of cancers[51]. However, using a soft agar assay, our previous work[2] showed that over-expressing cGAS may promote the oncogenic transformation of normal fibroblasts by suppressing HR repair. How can such a dis-crepancy be explained? In the soft agar experiment[2], normal cells rapidly replicated due to introduced vectors expressing large T antigen and Ras, which resulted in replication stress-induced DSBs

that required HR for repair. Overexpressing cGAS disrupted this repair process, thereby increasing the chance of tumorigenesis. However, in adults, most cells are in quiescent stages and HR repair is not triggered, suggesting that cGAS does not play an oncogenic role in these cells. In contrast, the L1 repression function of nuclear cGAS might be exerted throughout the cell cycle stages, thus cGAS may serve as a tumor suppressor in these quiescent somatic cells. More-over, it will be interesting to explore whether the double-faceted cGAS function is involved in the maintenance of genome integrity by suppressing HR and repressing L1 retrotransposition in other biolo-gical events, such as embryogenesis.

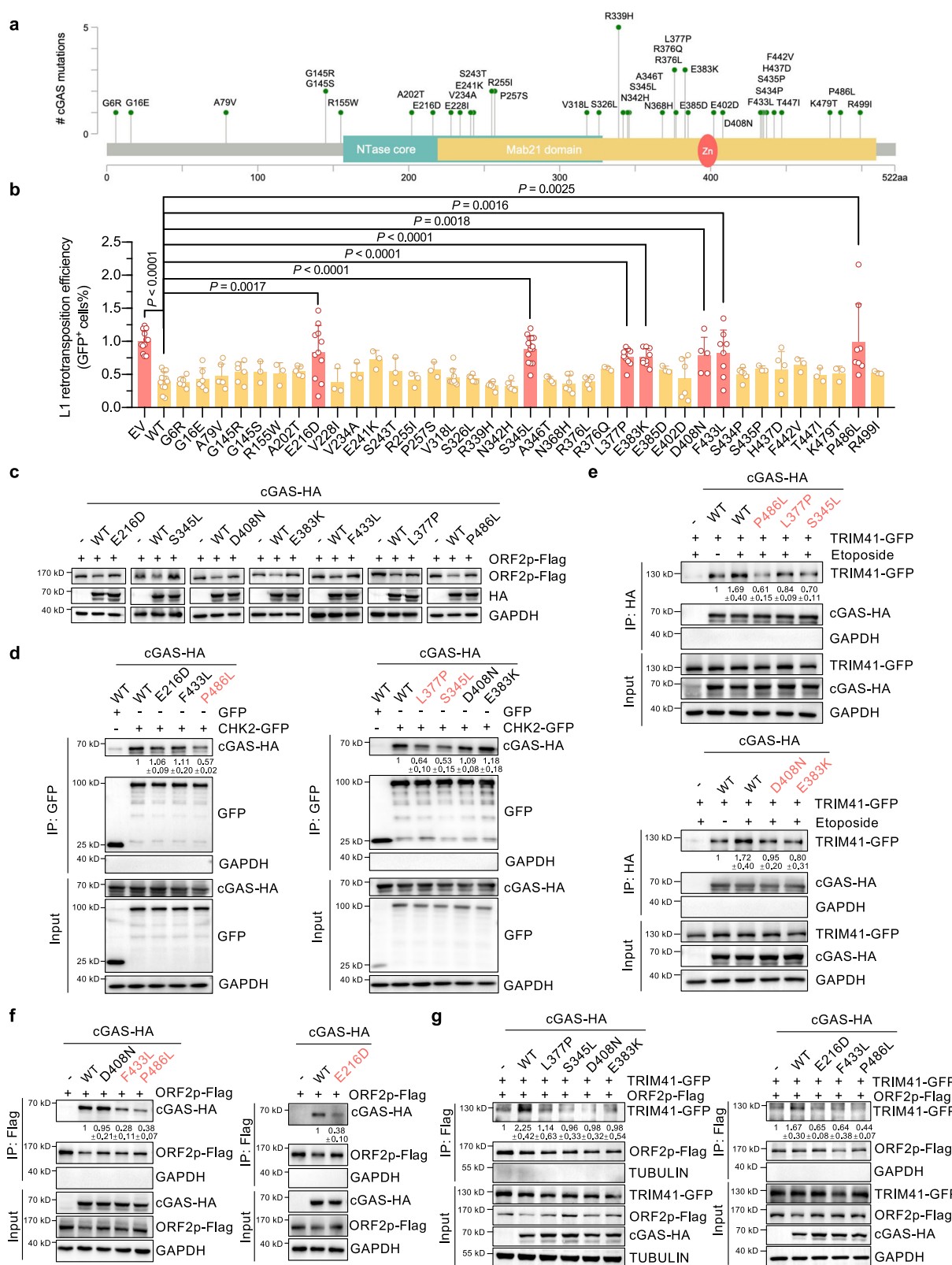

High levels of DNA damage can lead to deleterious outcomes, such as cellular and individual aging. Earlier studies have demonstrated that cGAS plays vital roles in regulating the onset of aging[52–54]. Moreover, several studies have indicated that L1 is actively transcribed in senescent cells and in several types of aging tissues[34,43,55–57]. The accumulation of cytoplasmic L1 cDNA activates cGAS to trigger a strong type I interferon response and sterile inflammation, which is a hallmark of aging[34]. Although the L1 mRNA level increases with the onset of aging, previous studies have indicated that cell division was required for L1 retrotransposition and that the L1 retrotransposition frequency was reduced in cells with cell cycle arrest[44,58], which is the fundamental feature of senescent cells. In this study, we demonstrated that in response to endogenous or exogenous DNA damage, CHK2 mediated cGAS phosphorylation at the S120 and S305 residues.

**Fig. 5 | Several cancer-associated cGAS mutations abolish the repressive effect of cGAS on L1 retrotransposition. a** Schematic showing cancer-associated cGAS mutations identified through data mining of the TCGA database. **b** The effect of overexpressing cGAS WT and cancer-associated cGAS mutants on L1 retrotransposition efficiency. Thirty-seven cancer-associated cGAS mutants were cloned and transfected into HeLa cells for the analysis of retrotransposition efficiency. Each dot represents an independent experiment, and $n$ = at least 3 independent experiments per group. Two-sided Student's $t$-test was performed. **c** Comparison of ORF2p protein levels with overexpressing cGAS WT or cancer-associated cGAS mutants. Cells were transfected with the indicated cGAS WT or cGAS mutants and ORF2p-Flag and harvested for use in Western blot analysis 24 h post-transfection. **d** Analysis of the interaction between CHK2 and WT or mutated cGAS. Cells were

transfected with the indicated plasmids and collected for co-IP experiments 24 h post-transfection. **e** Analysis of the association between TRIM41 and WT or mutated cGAS after DNA was damaged. The cells were transfected with the indicated plasmids and treated with 100 μg/mL etoposide for 90 min before being harvested for use in co-IP experiment 24 h post-transfection. **f** Analysis of the interaction between ORF2p and WT or mutated cGAS. Cells were transfected with the indicated plasmids and harvested for co-IP experiments 24 h post-transfection. **g** Analysis of the effect of overexpressing WT or mutated cGAS on the TRIM41-ORF2p interaction. Cells were transfected with the indicated plasmids and collected for co-IP experiments 24 h post-transfection. Data are presented as mean values ± s.d. Student's $t$ test was performed for (**b**). All the Inputs and IPs were from the same experiments. All experiments were repeated at least three times.

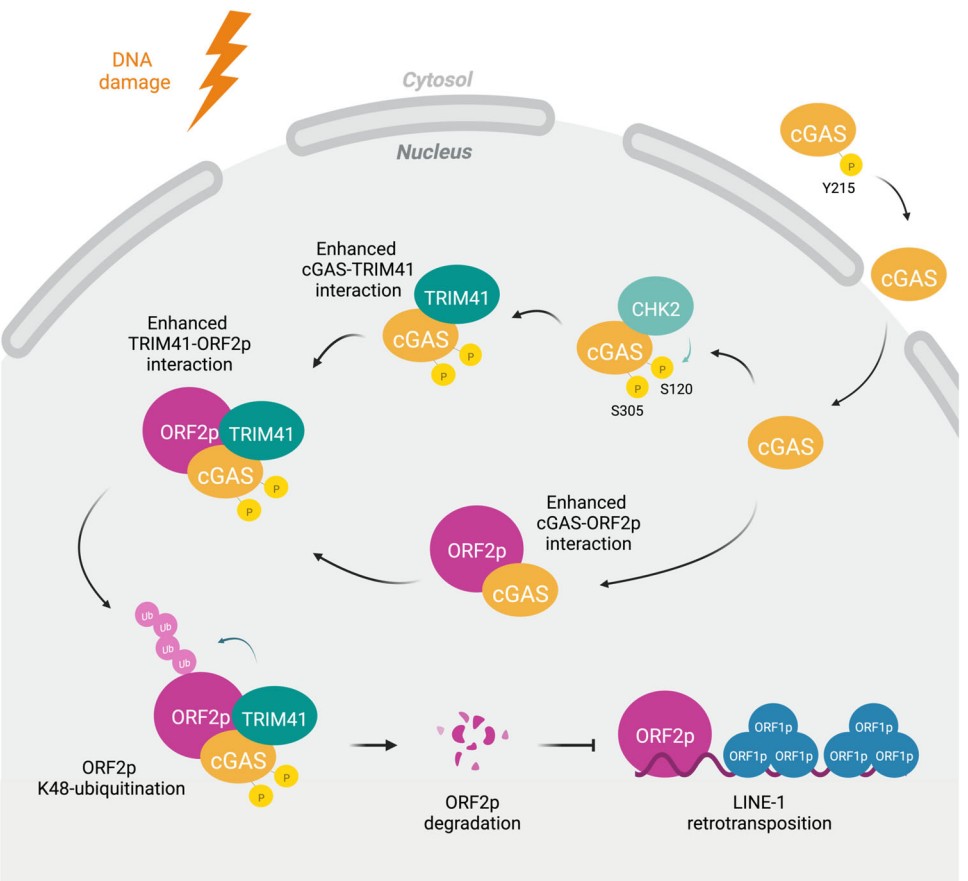

**Fig. 6 | Schematic diagram showing the regulatory mechanisms of nuclear cGAS-mediated repression on L1 retrotransposition.** Upon the occurrence of endogenous or exogenous DNA damage, cGAS is translocated into the nucleus and interacts with CHK2. CHK2-mediated phosphorylation of cGAS at Ser120 and Ser305 residues promotes the interaction between cGAS and the E3 ligase, TRIM41.

On the other hand, nuclear cGAS also interacts with ORF2p in a stress-dependent manner. The DNA damage-induced increase in cGAS-TRIM41 interaction, and cGAS-ORF2p interaction enhances TRIM41-ORF2p association, and results in the ubiquitination and degradation of ORF2p, thereby resulting in the repression of L1 retrotransposition. Created with BioRender.com.

Moreover, phosphorylation levels of these two residues are elevated in DNA damage-induced senescent cells, leading to suppressed L1 retrotransposition. Our data offer potential molecular insights into the seemingly contradictory outcomes of increased L1 mRNA and reduced retrotransposition in senescent cells and will contribute to the development of an antiaging approach involving inhibition of L1 retrotransposition.

Accumulating evidence has demonstrated that L1 is regulated at multiple stages in its retrotransposition cycle. At the transcriptional level, L1 is repressed by DNA methylation and specific histone modifications. Several factors, including MeCP2, KAP1, the NuRD complex, and sirtuins, are involved in regulating the accessibility of chromatin in L1 regions to influence L1 transcription[19,59–61]. At the translational level,

BRCA1 has been reported to suppress ORF2p translation in the cytoplasm by binding to L1 mRNA[62]. At the posttranslational modification stage, TEX19.1 and TREX1 have been shown to participate in the regulation of ORF1p protein degradation[31,32]. However, due to the relatively large molecular weight and low abundance of ORF2p, very little research has addressed the regulatory mechanism of ORF2p at the protein level. Whether ORF2p is regulated posttranslationally and the related mechanisms remain unclear. In this study, we identified TRIM41, an E3 ligase that ubiquitinates ORF2p and promotes its degradation. Moreover, we showed that cGAS promotes the TRIM41-ORF2p association to potentiate ORF2p degradation via proteasome pathway and repress L1 retrotransposition. Our work elucidates, for the first time, the mechanism of ORF2p posttranslational regulation. Nevertheless, other

types of posttranslational modifications of ORF2p and their related physiological significance warrant further investigation.

# Methods

## Research ethics

Our research complies with all relevant ethical regulations in Tongji University. All the animal studies were approved by the Biological Research Ethics Committee of Tongji University (TJAB04022103).

## Animals

The wild type and *Cgas* knock out mice (The Jackson Laboratory, stock no. 026554) were housed in a specific pathogen-free (SPF)-grade environment. Both 3–4-month-old male and female mice were used in this study.

## Cell culture

HCA2-hTERT and IMR90-hTERT[63] (IMRT90) cells were cultured in MEM (HyClone, Cat# SH30234) supplemented with 10% fetal bovine serum (Gibco), 1% nonessential amino acids (Gibco) and 1% penicillin–streptomycin (Gibco). HeLa cells and HEK293T cells were cultured in DMEM (Corning, Cat# 10-013-CVR) supplemented with 10% fetal bovine serum (Gibco) and 1% penicillin–streptomycin (Gibco). The cells were maintained in a 5% $CO_2$ humidified incubator (Thermo Fisher HERAcell 240i) at 37 °C. All the cells were periodically tested for free mycoplasma contamination via PCR.

## Reagents and plasmids

Puromycin (S9361), hygromycin (S2908) and olaparib (S1060) were purchased from Selleckchem. Etoposide (A1971), cycloheximide (A8244), MG132 (A2585) and inhibitors of ATM (A4605), ATR (B1383), CHK1 (A8638) and CHK2 (B1236) were purchased from APExBIO. The following antibodies were used: anti-Flag (MBL, Cat# M185-3L, 1/2000 for WB), Anti-Flag (Sigma–Aldrich, Cat. #F3165, 1/300 for PLA), anti-cGAS (Cell Signaling Technology, Cat# 15102, 1/800 for WB), anti-cGAS (Cell Signaling Technology, Cat# 79978, 1/300 for PLA), anti-β-tubulin (Bioworld, Cat# AP0064, 1/5000), anti-HA tag (Cell Signaling Technology, Cat# 3724, 1/1000), anti-GFP (ABclonal, Cat# AE012, 1/2000), anti-phospho-cGAS-S305 (ABclonal, Cat# AP1176, 1/1000), anti-phospho-cGAS-S120 (ABclonal, customized, 1/1000), anti-CHK2 (Cell Signaling Technology, Cat# 2662, 1/1000 for WB), anti-CHK2 (Cell Signaling Technology, Cat# 3440, 1/300 for PLA), anti-phospho-CHK2 (Cell Signaling Technology, Cat# 2661, 1/1000), anti-γH2AX (S139) (Cell Signaling Technology, Cat# 9718, 1/2000 for WB, 1/300 for IF), anti-H2AX (ABclonal, Cat# A11540, 1/2000), anti-Lamin A/C (ABclonal, Cat# A0249, 1/1000), anti-Ki67 (Thermo Fisher, Cat# MA5-14520, 1/500), anti-GAPDH (Proteintech, Cat# 60004, 1/5000), anti-rabbit IgG-HRP (Bio–Rad, Cat# 170-6515), anti-rabbit IgG H&L DyLight 488 (Abcam, Cat# ab96899, 1/500) and anti-mouse IgG H&L DyLight 594 (Abcam, Cat# ab96881, 1/500). The full-length coding sequences (CDSs) of E3 ligases (PIAS1, TRIM21, TRIM25, TRIM41, and ZNF598) were amplified with PCR using HeLa cDNA as the template and then cloned into an HA-tagged vector. The CDSs of L1 ORF1p and ORF2p were amplified with PCR using HeLa cDNA as the template and then cloned into a CMV promoter-driven, Flag-tagged or HA-tagged vector. The corresponding cancer-associated cGAS mutants were generated by site-directed mutagenesis based on the WT CDS of human cGAS.

Short interfering RNA (siRNA), short hairpin RNA (shRNA) and single guide RNA (sgRNA) sequences are listed in Supplementary Table 1.

## L1 retrotransposition assay

The retrotransposition efficiency was analyzed according to a previously reported protocol with some modifications[35]. Briefly, HeLa cells were plated in 6-well plates at a density of $2 \times 10^4$ cells per well. On Day 2 of culture, cells were cotransfected with 1 µg of the pEGFP-LRE3

vector, 100 ng of DsRed and the indicated amounts of vectors encoding cGAS or its mutants with the CN114 program on a Lonza 4D instrument. The medium was replaced with fresh medium containing 3 µg/mL puromycin twice, once on Day 5 and once on Day 7. Then, on Day 9, the surviving cells were harvested for flow cytometry analysis (BD Biosciences). To analyze the effect of etoposide on L1 retrotransposition efficiency, cells were treated with DMSO or 10 µg/mL etoposide for 20 min on Day 4. Then, the cells were treated with 3 µg/mL puromycin twice, on Day 6 and on Day 8. On Day 10, the surviving cells were harvested for flow cytometry analysis (BD Biosciences). The percentage of GFP-positive cells was determined with FlowJo software (Ashland, OR, USA).

## Comet assay

Cells were transfected with the indicated plasmids and collected for an alkaline comet assay the next day following the instructions of the comet assay kit manufacturer (Trevigen, Cat# 4250-050). The tail moment was analyzed with Cometscore software (Casplab 1.2.3b2). At least 50 cells were analyzed for each group.

## RNA extraction, genomic DNA extraction and real-time quantitative PCR

Total RNA was extracted using an RNAsimple Total RNA Kit (TIANGEN). One microgram of RNA was used for reverse transcription with a TransScript II Reverse Transcriptase kit (TRANS). For L1 copy number quantification, genomic DNA was isolated from indicated cells or tissues using a TIANamp Genomic DNA Kit (TIANGEN). qPCR was performed as described previously[37] Briefly, L1 genomic DNA content was probed using primers targeting ORF2 sequence, and the value was normalized to 5S ribosomal DNA content as the measurement of relative L1 copy number. Real-time quantitative PCR was performed with FastStart Universal SYBR Green Master Mix (Roche) on a Vii7 real-time PCR machine (Life Technologies). The indicated primers are listed in the Supplementary Table 2.

## Coimmunoprecipitation

Cells transfected with the indicated plasmids were collected 24 h post-transfection. Then, the cells were lysed in lysis buffer (20 mM HEPES (pH 8.0), 150 mM NaCl, 0.2 mM EDTA, 1% NP-40 and 5% glycerol) supplemented with protease cocktail for 30 min on ice. The cells were then sonicated at 10% power for 3 min at 2-s intervals, followed by centrifugation at $13,500 \times g$ for 15 min. The supernatants were incubated with antibody-conjugated beads at 4 °C overnight and washed four times with ice-cold lysis buffer the next day. The beads were resuspended in 2× sample buffer and boiled for 10 min prior to Western blot analysis. Immunoblot images generated using fluorescence antibodies were converted to grayscale in the main figures.

## Coimmunoprecipitation–coupled mass spectrometry analysis

To explore the potential targets interacting with ORF2p, HEK293T cells were transfected with plasmids encoding Flag-tagged ORF2p, and these cells were collected and lysed with lysis buffer 24 h post-transfection. The cells were then sonicated and centrifuged, and the supernatants were used for immunoprecipitation with anti-Flag antibody-conjugated beads. All precipitated samples were used for a mass spectrometry analysis.

## Stable isotope labeling by amino acids in cell culture (SILAC)

The SILAC experiment was conducted using a SILAC-DMEM (K6R6) Kit (imultiomics, Cat# SM202101) following the manufacturer's instructions. Briefly, sgCtrl cells were cultured in the medium containing normal Lysine (K0) and Arginine (R0), while sgcGAS cells were maintained in $^{13}C_6$-Lysine (K6) and $^{13}C_6$-Arginine (R6) medium, for 7 passages in parallel, before being transfected with plasmids encoding

Flag-tagged ORF2p. At 24 h post transfection, an equal number of cells were lysed with lysis buffer (20 mM HEPES (pH 8.0), 150 mM NaCl, 0.2 mM EDTA, 1% NP-40 and 5% glycerol) and sonicated. After centrifugation, the supernatant was subjected to immunoprecipitation using an excess amount of anti-Flag conjugated beads. Finally, equal amounts of precipitated samples from both groups were mixed, and subjected to mass spectrometry analysis.

### Generation of cGAS- and TRIM41-knockout cells using the CRISPR–Cas9 system

Oligos targeting human cGAS or TRIM41 or CHK2 were synthesized and ligated into CRISPR-PX458 vectors and then transfected into HeLa cells. GFP$^+$ cells were sorted 24 h post-transfection on a BD FACS AriaII machine. Successfully targeted cells were confirmed by Western blotting using antibodies against cGAS and TRIM41 and by Sanger sequencing of the PCR products of the flanking region of the targeted sgRNA sequences. The indicated oligonucleotide sequences are listed in the Supplementary Table 1.

### Protein purification

The CDS of cGAS was amplified and inserted into the pGEX-4T vector and then transformed into the *E. coli* BL21 strain. Bacteria were cultured at 37 °C in 2 × YT medium supplemented with 100 μg/mL ampicillin for 4 h. Once the concentration of cultures reached an OD600 of 0.6–0.8, 0.2 mM isopropyl-beta-D-thiogalactopyranoside (IPTG) was added to the medium and cultured overnight at 16 °C. The next day, the bacteria were collected and lysed with GST lysis buffer (50 mM Tris (pH 7.5), 150 mM NaCl, 0.05% NP-40 and 200 μM PMSF) supplemented with 5 mg/mL lysozyme on ice for 60 min. The cell lysates were sonicated, and cell debris was removed by centrifugation at $13,500 \times g$ for 15 min. Then, 100 μL of glutathione agarose was added to the supernatants and rotated at 4 °C overnight. After washing with GST lysis buffer six times and with 100 mM Tris (pH 8.0) twice, GST-cGAS was released from the Sepharose beads with elution buffer (100 mM Tris (pH 8.0) and 20 mM glutathione) for 1 h at 4 °C. To purify the TRIM41 proteins, HEK293T cells were transfected with a plasmid encoding ORF2p-Flag and purified with anti-Flag beads. Then, the proteins were released from Flag antibody-conjugated beads with elution buffer (0.2 M glycine (pH 2.5)), and the supernatants were neutralized with 1 M Tris-HCl (pH 10.4). TRIM41-HA and cGAS-GFP were purified using the same procedures with HA antibody-conjugated beads and GFP traps.

### In vitro co-IP assay

Purified proteins were incubated in in vitro co-IP buffer (25 mM Tris-Cl, 150 mM NaCl, 1% NP-40). Then, antibody (or antibody-conjugated beads) was added, and the mixture was incubated on a rotator at 4 °C overnight. Protein A/G-agarose beads were added to the mixture for incubation at 4 °C for 2 h the next day when non-conjugated primary antibody was used for immunoprecipitation. Subsequently, the pellets were washed 5 times with lysis buffer and boiled for 10 min prior to Western blot analysis.

### Immunofluorescence staining

Cells transfected with the indicated plasmids were seeded on coverslips in 12-well plates. Twenty hours post-transfection, the cells were rinsed with cold PBS and fixed with 4% paraformaldehyde (PFA) for 15 min at room temperature. Then, the cells were permeabilized with 0.25% Triton X-100 for 30 min and blocked with 2% bovine serum albumin (BSA) for 1 h at room temperature. The cells were then incubated with primary antibody at 4 °C overnight and with secondary antibody at room temperature for 1 h in the dark the following day. Finally, the cells were mounted in mounting medium with DAPI. Images were acquired with a Nikon laser scanning confocal microscope.

### Cell fractionation

Cells transfected with the indicated plasmids were lysed in hypotonic buffer (2 mM MgCl$_2$, 20 mM Tris (pH 7.6), 50 mM 2-mercaptoethanol, 0.1 mM EDTA, 1 mM PMSF) for 5 min at 4 °C 24 h post-transfection. Then, the cell lysates were centrifuged at $200 \times g$ for 5 min, and the supernatants (cytoplasmic fraction) were collected. The precipitates were resuspended in hypertonic buffer (20 mM HEPES (pH 7.9), 400 mM NaCl, 1 mM glycerol, 1 mM PMSF, 0.5 mM NaF, 0.5 mM Na$_3$VO$_4$, 0.5 mM DTT) and lysed at 4 °C for 20 min, followed by centrifugation at $12,000 \times g$ for 15 min to remove the nuclear debris. Finally, the supernatants (nuclear fraction) were collected. The fractions were directly used for Western blot analysis or for co-IP analysis.

### Proximity ligation assay (PLA)

HeLa cells were seeded on coverslips. At 24 h post seeding, cells were washed twice with cold PBS and fixed with 4% paraformaldehyde at room temperature for 15 min. Subsequently, cells were washed twice with PBS and permeabilized with 0.5% Triton X-100 at room temperature for 10 min. Cells were then washed with PBS three times. The following experimental procedures were performed according to the manufacturer's instructions of a Duolink In Situ Red Starter Kit (Sigma, Cat# DUO92101).

### In vitro kinase assay

The assay was performed according to previously reported protocols[11]. Briefly, purified HA-tagged wild-type CHK2, CHK2-T68D and CHK2-D347A mutants were mixed with GST-cGAS in ice-cold kinase buffer (50 mM Tris (pH 7.5), 2 mM DTT, 1 mM β-glycerophosphate, 10 mM MgCl$_2$, 10 μM ATP, 0.1 mM Na$_3$VO$_4$). The mixtures were incubated at 30 °C for 1 h prior to being boiled for Western blot analysis.

### Generation of an anti-phospho-cGAS S120 antibody

Rabbit polyclonal antibody was raised against cGAS phosphorylated S120 (p-cGAS-S120) in collaboration with ABclonal Biotech. In brief, 3 rabbits were immunized with the peptide c(KLH)[116]SRAGS(p-S)CRQR[124], in which c(KLH) indicates keyhole limpet hemocyanin fused at a cysteine residue, and p-S indicates phosphorylated serine. The unphosphorylated peptide c(KLH)[116]SRAGSCRQR[124] was used as a control during antibody purification and detection.

### Statistics and reproducibility

No statistical methods were used to predetermine sample size. Sample sizes were estimated based on experiences on the similar experiments performed by us and other published studies. Methods of statistical analysis employed to determine the significance of difference are reported in related figure legends, and the exact *P* values are indicated in figures. Statistical calculations were performed using the Prism software (GraphPad).

### Reporting summary

Further information on research design is available in the Nature Portfolio Reporting Summary linked to this article.

## Data availability

The data supporting the conclusions in this manuscript can be found in the main text or the supplementary materials and are available from the corresponding author upon request. Source data are provided with this paper.

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

## Acknowledgements

This work was supported by the National Key R&D Program of China (2022YFA1103703 and 2021YFA1102003 to Z.M.), the National Natural Science Foundation of China (Grant Nos. 82225017, 32270750 and 82071565 to Z.M., 81972457 and 32171288 to Y.J., 32200595 to Y.C.), the Shanghai Sailing Program (22YF1434300 to Y.C.).

## Author contributions

Z.M. and Y.J. conceived the project, designed experiments, and supervised the study. Z.Z., Y.C., H.W., H.T., and H.Z. performed experiments with the help from H.L. Z.Z. and Y.C. contributed to data analysis and interpretation. The paper was written by Z.M., Y.C., and Z.Z.

## Competing interests

The authors declare no competing interests.
