## [Peer Review File · Nature Communications]

Nuclear cGAS restricts L1 retrotransposition by promoting TRIM41-mediated ORF2p ubiquitination and degradationEditorial Note: Parts of this Peer Review File have been redacted as indicated to maintain the confidentiality of unpublished data.

REVIEWER COMMENTS

Reviewer #1 (Remarks to the Author):

Questions surrounding the role of cGas and LINE1 in inflammation and other regulatory processes in the cell have picked up significant traction in recent years. The manuscript presented here by Zhen et. al proposes a novel regulatory role cGas may play against DNA damage, L1 retrotransposition, and cancer. The results of this paper are quite promising, however, more experiments (in addition to including experiments described but not shown) are needed to satisfy the claims made and the language used to describe the work, in addition to a handful of misleading or incorrect statements, needs to be amended before this work can be considered for publication.

In line 50 of the manuscript, the authors state that 'cGas mainly resides in the cytosol' when this has been shown to not be the case (1). Although cGas has been shown to be most active in the cytosol thus far, a majority of cGas actually resides in the nucleus. Additionally, the authors state in line 54 that necessity of cGas in the nucleus has yet to be determined, when it has been published that nuclear cGas is necessary for an inflammatory response to occur in response to nucleic acids in the cytoplasm (2). In line 80, the authors state that cells have evolved mechanisms to prevent cGas from binding to nuclear DNA, when it is actually has been shown to bind to nuclear chromatin; what is not known is how cGas is able to differentiate between this DNA and foreign, pathogenic DNA (3).

The authors discuss at length a soft agar assay in the discussion (lines 341-352) that is not shown in any figure or cited elsewhere. If the authors wish to discuss this work, it should be accessible to the reader. In line with this soft agar experiment, much of the discussion appears to not stay on track and is disorganized.

Regarding the experiments conducted for this paper, much of the design is rational. However, the authors rely very heavily on western blotting, and this method is used to prove most of their findings. Diversification of experimental results would greatly bolster the claims made by this manuscript. For example, figure one states that cGas prohibits L1 retrotransposition by reducing the levels of ORF2p. This can be further proved by a simple SILAC experiment followed by mass spectrometry. Same for cGAS-ORF2-TRIM41 interaction.

The authors also include qPCR data showing a loss of cGas reduces L1 mRNA levels. How is this not also contributing to the reduction of ORF2p? The authors also maintain that there is no reduction of L1 mRNA between cell types despite their own evidence showing otherwise.

1. Wu Y, Song K, Hao W, Li J, Wang L, Li S. Nuclear soluble cGAS senses double-stranded DNA virus infection. *Commun Biol.* 2022
2. Sun H, Huang Y, Mei S, et al. A Nuclear Export Signal Is Required for cGAS to Sense Cytosolic DNA. *Cell Rep.* 2021
3. Li T, Huang T, Du M, et al. Phosphorylation and chromatin tethering prevent cGAS activation during mitosis. *Science.* 2021

Reviewer #2 (Remarks to the Author):

In this study, Zhen et al. found that nuclear cGAS inhibit LINE-1 retrotransposition. Mechanistically, they revealed DNA damage induced nuclear cGAS phosphorylated by CHK2, which promoted the cGAS and E3 ligase TRIM41 interaction with ORF2p, thereby facilitating TRIM41-mediated ORF2p K48 ubiquitination and degradation. Previously, the author's team found that nuclear cGAS suppresses DNA HR repair. It is interesting that this study they expand the role of cGAS in maintaining gene stability. The LINE-1 retrotransposition restrictive function of cGAS is also divided from the cytosolic DNA sensor function. These findings are interesting and novel, and most of the data are convincing. Here are the comments for this article:

Major comment:

1. The inhibitory effect of cGAS on LINE-1 retrotransposition was happened without external DNA damage stimulation in Figure 1 and Figure 2. Do DNA damage increased ORF2 degradation and inhibition of LINE-1 retrotransposition?
2. Many of the biochemical assays were done using overexpressed protein. It would be better to do some assays using endogenous ORF2.
3. Does CHK2-phosphorylation of ORF2 affect its nuclear localization?
3. What is the localization of TRIM41? Does DNA damage treatment increase the TRIM41 in the nuclear and promote the interaction of cGAS and TRIM41?

Minor comment:

1. The references start with number 4 (line 73) should be checked.
2. The quality of Figure 3b lamin A blot is not high enough.

Reviewer #3 (Remarks to the Author):

In their manuscript entitled "Nuclear cGAS restricts L1 retrotransposition by promoting TRIM41-mediated ORF2p ubiquitination and degradation", Zhen et al. present evidence for a function of cGAS in preventing LINE-1 retrotransposition by mediating the degradation of ORF2p via the TRIM41 ubiquitin ligase. Overall, the experiments argue for the veracity of this finding. However, a number of items need to be addressed, such as the lack of relevant controls on the IPs, the large reliance on non-quantitative data, and the fact that it is not clear if the authors' findings hold relevance for endogenous LINE-1, given that overexpression setups were used throughout. These points are described in more detail below.

Major points

1. Lack of controls in the immunoprecipitation experiments. Essentially none of the IPs contain a negative control (ie, a protein that should not be pulled down, for example, tubulin, actin etc.). Therefore the results cannot properly be assessed. Please add this for all the IPs.
2. How is cGAS impacting L1 transcription, ORF2p stability and L1 retrotransposition under normal conditions? The authors only use artificial L1 reporters, ORF2p overexpression systems etc., but the impact on endogenous proteins and mechanisms is not assessed. This could for example be addressed with the retrotransposition increase that seems to occur in senescent cells (see for example PMID 30728521).
3. Some very detailed conclusions are made from non-quantitative data. Most of the time this is IP of overexpressed proteins, and ubiquitin blots. These should be addressed by repeats of the IPs with quantifications using fluorescent antibodies, and affinity measurements using the recombinant proteins. The ubiquitination blots in particular need some repeats/statistics.
4. The phospho-specific antibodies. The authors need to provide proof that these work as expected (WB with wt/mutant proteins, peptides, peptide-competitions experiments etc.).

Minor points

1. Lines 141-145: The impact of the mutations interacting with DNA are potentially overinterpreted. As observed in PMID 31299200, a mutant more severe than the one used here only reduced affinity for DNA by two-fold. This should be rewritten.
2. Fig. 1g, h, l, m: It is strange that loss of cGAS results in less production of the L1 reporter. This should be discussed.
3. Fig. 1u, part of the label seems to be missing.
4. With regards to Fig. 2a, b, the authors should also test the flipside – protein stability, by treating cells with cyclohexamide in order to shut down fresh Orf2 production.
5. The mutations that seem to impact interaction with CHK2 are all over cGAS when analysed on cGAS structures. This should be interpreted/discussed in the discussion.
6. Extended Data Fig. 4d seems to not match the text.
7. In Fig. 4c, HA-cGAS pulls down less Orf2 if it is L377P, but in Ext Fig. 4 ORF2 can pull down just as much L377P cGAS as wt. Can this be explained/discussed?
8. Please add MW labels to all WBs.

REVIEWER COMMENTS

Reviewer #1 (Remarks to the Author):

Questions surrounding the role of cGas and LINE1 in inflammation and other regulatory processes in the cell have picked up significant traction in recent years. The manuscript presented here by Zhen et. al proposes a novel regulatory role cGas may play against DNA damage, L1 retrotransposition, and cancer. The results of this paper are quite promising, however, more experiments (in addition to including experiments described but not shown) are needed to satisfy the claims made and the language used to describe the work, in addition to a handful of misleading or incorrect statements, needs to be amended before this work can be considered for publication.

We thank the reviewer for the positive comments.

In line 50 of the manuscript, the authors state that ‘cGas mainly resides in the cytosol’ when this has been shown to not be the case (1). Although cGas has been shown to be most active in the cytosol thus far, a majority of cGas actually resides in the nucleus. Additionally, the authors state in line 54 that necessity of cGas in the nucleus has yet to be determined, when it has been published that nuclear cGas is necessary for an inflammatory response to occur in response to nucleic acids in the cytoplasm (2). In line 80, the authors state that cells have evolved mechanisms to prevent cGas from binding to nuclear DNA, when it is actually has been shown to bind to nuclear chromatin; what is not known is how cGas is able to differentiate between this DNA and foreign, pathogenic DNA (3).

We thank the reviewer for pointing them out. As the reviewer suggested, we have rephrased these sentences in our revised manuscript.

The authors discuss at length a soft agar assay in the discussion (lines 341-352) that is not shown in any figure or cited elsewhere. If the authors wish to discuss this work, it should be accessible to the reader. In line with this soft agar experiment, much of the discussion appears to not stay on track and is disorganized.

We apologize for the confusion. We have reorganized the discussion section as the reviewer suggested.

Regarding the experiments conducted for this paper, much of the design is rational. However, the authors rely very heavily on western blotting, and this method is used to prove most of their findings. Diversification of experimental results would greatly bolster the claims made by this manuscript. For example, figure one states that cGas prohibits L1 retrotransposition by reducing the levels of ORF2p. This can be further proved by a simple SILAC experiment followed by mass spectrometry. Same for cGAS-ORF2-TRIM41 interaction.

We thank the reviewer for the valuable suggestion, and we have performed additional experiments to strengthen our conclusions. To further validate that cGAS regulates the protein level of ORF2p, a SILAC-based quantitative mass spectrometry assay was performed as the reviewer suggested. Consistent with the Western blotting result, the mass spectrometry data showed that cGAS knockout significantly increased the protein level of ORF2p by 10.48-fold in HeLa cells. To study whether cGAS influences the protein interaction between ORF2p and TRIM41, a proximity ligation assay (PLA) was carried out. In line with the Western blotting result, the PLA data confirmed that cGAS overexpression significantly promoted the association between ORF2p and TRIM41.

Fig. 1i

	Accession	Abundance: light (sgCtrl)	Abundance: heavy (sgcGAS)	Ratio: (sgcGAS/sgCtrl)
Repeat 1	O00370	254799.625	1616217.750	6.34
Repeat 2	O00370	757408.281	8019825.875	10.59
Repeat 3	O00370	819453.000	11900879.781	14.52

Supplementary Fig. 2j

The authors also include qPCR data showing a loss of cGAS reduces L1 mRNA levels. How is this not also contributing to the reduction of ORF2p? The authors also maintain that there is no reduction of L1 mRNA between cell types despite their own evidence showing otherwise.

We thank the reviewer for raising the intriguing question. As the reviewer mentioned, depletion of cGAS reduced L1 mRNA levels. However, it did not result in an expected reduction in ORF2p protein level. On the contrary, the ORF2p protein level significantly increased upon the loss of cGAS. These seemingly contradictory results indicated that (1) the cGAS-mediated suppression in ORF2p protein level was not through transcriptional mechanisms; (2) there might exist other layers of regulation on L1 mRNA level, such as a negative feedback of ORF2p protein on L1 transcription, and a potential influence of cGAS on L1 5' UTR promoter activity and mRNA stability. These observations raise fascinating avenues for future exploration. We have rephrased the sentence, and added more discussion on this point in our revised manuscript.

1. Wu Y, Song K, Hao W, Li J, Wang L, Li S. Nuclear soluble cGAS senses double-

stranded DNA virus infection. *Commun Biol.* 2022

2. Sun H, Huang Y, Mei S, et al. A Nuclear Export Signal Is Required for cGAS to Sense Cytosolic DNA. *Cell Rep.* 2021

3. Li T, Huang T, Du M, et al. Phosphorylation and chromatin tethering prevent cGAS activation during mitosis. *Science.* 2021

Reviewer #2 (Remarks to the Author):

In this study, Zhen et al. found that nuclear cGAS inhibit LINE-1 retrotransposition. Mechanistically, they revealed DNA damage induced nuclear cGAS phosphorylated by CHK2, which promoted the cGAS and E3 ligase TRIM41 interaction with ORF2p, thereby facilitating TRIM41-mediated ORF2p K48 ubiquitination and degradation. Previously, the author's team found that nuclear cGAS suppresses DNA HR repair. It is interesting that this study they expand the role of cGAS in maintaining gene stability. The LINE-1 retrotransposition restrictive function of cGAS is also divided from the cytosolic DNA sensor function. These findings are interesting and novel, and most of the data are convincing. Here are the comments for this article:

We thank the reviewer a lot for the positive comments.

Major comment:

1. The inhibitory effect of cGAS on LINE-1 retrotransposition was happened without external DNA damage stimulation in Figure 1 and Figure 2. Do DNA damage increased ORF2 degradation and inhibition of LINE-1 retrotransposition?

We thank the reviewer for raising the intriguing question. To investigate the impact of external DNA damage stimulation on cGAS-mediated LINE-1 inhibition, DNA damage was induced in HeLa cells with etoposide, a topoisomerase II inhibitor. The results suggested that DNA damage significantly potentiated the inhibitory effect of cGAS on ORF2p protein level, and LINE-1 retrotransposition efficiency. The data is shown below.

2. Many of the biochemical assays were done using overexpressed protein. It would be better to do some assays using endogenous ORF2.

We thank the reviewer very much for the suggestion. We fully agree with the reviewer

that it makes more sense to study the regulatory effect of cGAS on endogenous ORF2p. However, a number of studies have indicated that ORF2p translated at a low level (Taylor et al, *Cell*, 2013; Alisch et al, *Genes & Development*, 2006), making it extremely difficult to directly study endogenous ORF2p. Although a few studies claimed that they have successfully developed ORF2p antibodies, actually all of these antibodies could detect exogenously overexpressed recombinant ORF2p while failing to detect endogenous ones (Ardeljan et al, *Nature Structural & Molecular Biology*, 2020; Ardeljan et al, *Mobile DNA*, 2020; Sokolowski et al, *Mobile DNA*, 2014). Therefore, it is widely accepted to overexpress exogenous ORF2p to increase its production to facilitate the mechanistic studies (Miyoshi et al, *Molecular Cell*, 2019; Boeke et al, *Nature Structural & Molecular Biology*, 2020).

Alternatively, a previously reported assay to directly analyze genomic LINE1 content was performed to further confirm our findings, and we found that the relative LINE1 DNA level was significantly increased in cGAS knockout HeLa cells and in the brain and kidney of *Cgas* knockout mice. These data are shown as below and have also been included in our revised manuscript.

Supplementary Fig. 1 d to f

3. Does CHK2-phosphorylation of ORF2 affect its nuclear localization?

We thank the reviewer for the comments, and apologize for any confusion. We have demonstrated that CHK2 phosphorylated cGAS at both the S120 and S305 residues in our previous version of manuscript, and we did not report any phosphorylation modification of ORF2p mediated by CHK2.

To further investigate whether CHK2-mediated phosphorylation of cGAS affects its subcellular localization, HA-tagged wildtype cGAS, cGAS S120A/S305A mutant, and S120E/S305E mutant were introduced into HeLa cells. After etoposide treatment, immunofluorescence staining was performed with anti-HA, and no significant difference on cGAS localization was observed. Moreover, treating cells with the CHK2 inhibitor, BML-277, did not significantly alter cGAS nuclear localization, either. Therefore, we concluded that CHK2-mediated cGAS phosphorylation did not influence the nuclear localization of cGAS. These data are shown below and have also been included in our revised manuscript.

Supplementary Fig. 4i

4. What is the localization of TRIM41? Does DNA damage treatment increase the TRIM41 in the nuclear and promote the interaction of cGAS and TRIM41?

We thank the reviewer for the comments. Subcellular fractionation experiments were performed, and our data showed that endogenous TRIM41 mainly localized in the nucleus in both HeLa cells and HEK293T cells, without external DNA damage stimulation. To investigate whether DNA damage enhanced the interaction between cGAS and TRIM41 by altering TRIM41 localization, both HeLa cells and HEK293T cells were treated with etoposide, however, DNA damage did not significantly impact the nuclear localization of TRIM41. This data is shown below and also included in our revised manuscript.

Supplementary Fig. 4h

Minor comment:

1. The references start with number 4 (line 73) should be checked.

We thank the reviewer for pointing it out. We have made the corrections.

2. The quality of Figure 3b lamin A blot is not high enough.

We thank the reviewer for pointing it out. We have provided another Lamin A/C blot of higher quality.

Fig. 3b

Reviewer #3 (Remarks to the Author):

In their manuscript entitled “Nuclear cGAS restricts L1 retrotransposition by promoting TRIM41-mediated ORF2p ubiquitination and degradation”, Zhen et al. present evidence for a function of cGAS in preventing LINE-1 retrotransposition by mediating the degradation of ORF2p via the TRIM41 ubiquitin ligase. Overall, the experiments argue for the veracity of this finding. However, a number of items need to be addressed, such as the lack of relevant controls on the IPs, the large reliance on non-quantitative data, and the fact that it is not clear if the authors’ findings hold relevance for endogenous LINE-1, given that overexpression setups were used throughout. These points are described in more detail below.

We thank the reviewer for the positive comments and critical suggestions.

Major points

1. Lack of controls in the immunoprecipitation experiments. Essentially none of the IPs contain a negative control (ie, a protein that should not be pulled down, for example, tubulin, actin etc.). Therefore the results cannot properly be assessed. Please add this for all the IPs.

We thank the reviewer for the suggestion. Actually, all the IPs were performed with either GAPDH or TUBULIN as a negative control in our study. However, these negative controls were not included in our previous version of figures. Some representative negative control images for the key results are shown as below and also included in our revised manuscript. To ensure the figures are not too crowded and remain easily readable, we have to present the negative controls for other IPs in the Supplementary figures as uncropped western blot images. But if the reviewer or editor feels it necessary to include them in the figures, we would be happy to follow the reviewer or editor’s instruction.

Fig. 2c

Fig. 2d

Fig. 2h

Fig. 2j

Fig. 2k

Fig. 2l

Fig. 3e

Fig. 3f

Fig. 3g

2. How is cGAS impacting L1 transcription, ORF2p stability and L1 retrotransposition under normal conditions? The authors only use artificial L1 reporters, ORF2p overexpression systems etc., but the impact on endogenous proteins and mechanisms is not assessed. This could for example be addressed with the retrotransposition increase that seems to occur in senescent cells (see for example PMID 30728521).

We thank the reviewer for the suggestion, and we agree with the reviewer that it is important to investigate the impact of cGAS on endogenous L1.

(1) We have shown that cGAS reduced ORF2p protein level at the post-translational stage. To test whether the inhibitory effect was also partly mediated by transcriptional regulation, as the reviewer mentioned, qPCR was performed with a pair of L1 specific primers as previously reported (De Cecco et al, *Nature*, 2019). The results indicated that cGAS knockdown or knockout did not lead to an increase of L1 mRNA level, ruling out the possibility that cGAS transcriptionally repressed ORF2p level. On the contrary, we observed an intriguing phenomenon, that depletion of cGAS led to a significant decrease in L1 mRNA level, implying that there existed other aspects of cGAS-mediated regulation of L1, such as L1 transcription. As we shown in Fig.1g to h, loss of cGAS diminished the promoter activity of L1 5'UTR, thus we hypothesized that cGAS or ORF2p *per se* may also regulate L1 mRNA transcription. In addition, there may also exist a potential influence of cGAS on L1 mRNA stability. However, we believed that this is beyond the scope of our study, and we are pleased to figure these issues out in our future work. The data have been included in Fig. 1l and m.

(2) The reviewer also suggested us to explore the influence of cGAS on endogenous ORF2p. As we replied to Reviewer #2, ORF2p translated at a low level (Taylor et al, *Cell*, 2013; Alisch et al, *Genes & Development*, 2006), thus it is extremely difficult to directly study endogenous ORF2p. Moreover, although several studies claimed that they have successfully developed specific antibodies against ORF2p, actually none of these antibodies did detect endogenous ORF2p (Ardeljan et al, *Nature Structural & Molecular Biology*, 2020; Ardeljan et al, *Mobile DNA*, 2020; Sokolowski et al, *Mobile DNA*, 2014). Therefore, overexpressing exogenous ORF2p to increase its production is

a widely accepted manner for its mechanistic studies (Miyoshi et al, *Molecular Cell*, 2019; Boeke et al, *Nature Structural & Molecular Biology*, 2020).

(3) To study whether cGAS regulated L1 insertions, a previously reported qPCR-based method for analyzing the endogenous L1 copy number (Coufal et al., *Nature*, 2010) was carried out. By using this method, we first explored whether cGAS knockout affects L1 copy number in HeLa cells. A pair of L1 specific primers were used for amplifying L1 DNA, and the results confirmed that depletion of cGAS significantly increased genomic L1 DNA content, indicating that retrotransposition events occur more frequently in cGAS knockout cells, compared to the wildtype cells. In addition, we also compared L1 DNA content between wildtype and *Cgas* knockout mice. The data clearly demonstrated that both the organs of kidney and brain had elevated genomic L1 DNA content in *Cgas* knockout mice, relative to wildtype mice, suggesting a conserved function of cGAS on L1 retrotransposition between human and mice. The data are shown as below and also included in the Supplementary Fig. 1d, e and f in our revised manuscript. Collectively, these additional experiments, without artificial L1 reporters or ORF2p overexpression systems, further strengthened the physiological significance of cGAS-mediated L1 repression under normal conditions.

Supplementary Fig. 1 d to f

(4) In addition, when we responded to the concerns that whether DNA damage induced cGAS-mediated ORF2p degradation and L1 retrotransposition inhibition, raised by Reviewer #2, we found that HeLa cells treated with etoposide actually entered a state of senescence, evidenced by upregulation of p21, an increase in the positive rate of β -gal staining, and an obvious growth arrest (shown in Supplementary Fig. 5a to e). As we mentioned above, L1 retrotransposition efficiency declined more robustly upon cGAS overexpression in this condition. Mechanistically, we found an increase in the phosphorylation levels of CHK2, and cGAS S120/S305 residues, in a panel of senescent cells, including HeLa, IMR90-hTERT and HCA2-hTERT. These data are shown as below and have also been included in Fig. 4a and Supplementary Fig. 5a to e. Our data provided a possible molecular explanation on the seemingly contradictory outcomes of increased L1 mRNA and reduced retrotransposition in senescent cells.

Fig. 4 a to f

3. Some very detailed conclusions are made from non-quantitative data. Most of the time this is IP of overexpressed proteins, and ubiquitin blots. These should be addressed by repeats of the IPs with quantifications using fluorescent antibodies, and affinity measurements using the recombinant proteins. The ubiquitination blots in particular need some repeats/statistics.

We thank the reviewer for the valuable suggestion. We have repeated these IPs (including all the ubiquitination blots, as the reviewer specifically pointed out, as well as the interactions between cGAS and CHK2 or its mutant, TRIM41 and cGAS or its mutants, and TRIM41 and ORF2p, etc.), and the quantification results have been shown in the corresponding panels (Fig. 2c, 2d, 2j, 2k, 2l, Supplementary Fig. 2i, 2k, 2l, Fig. 3g, 3l and 3m, Supplementary Fig. 4c, 4f, 4j, 4n, Fig. 5d, 5e, 5f and 5g). Moreover, as the reviewer suggested, we also repeated some IPs using fluorescent antibodies, and the results were similar to those generated with HRP-conjugated antibodies. These data are shown as below, and have also been included in our revised manuscript (Supplementary Fig. 3f, 4d, 4e, 4g, 4k, 4o, 5d, 5e, 5g and 5h). We appreciate the reviewer a lot for raising this issue to help us further strengthen our main conclusions.

Fig. 2c

Fig. 2d

Fig. 2j

Fig. 2k

Fig. 2l

Supplementary Fig. 2i

Supplementary Fig. 2k

Supplementary Fig. 2l

Fig. 3g

Fig. 3l

Fig. 3m

Supplementary Fig. 4c

Supplementary Fig. 4f

Supplementary Fig. 4j

Supplementary Fig. 4n

Fig. 5d

Fig. 5e

Fig. 5f

Fig. 5g

Supplementary Fig. 3f

Supplementary Fig. 4a

Supplementary Fig. 4d

Supplementary Fig. 4e

Supplementary Fig. 4g

Supplementary Fig. 4k

Supplementary Fig. 4o

Supplementary Fig. 5d

Supplementary Fig. 5e

Supplementary Fig. 5g

Supplementary Fig. 5h

4. The phospho-specific antibodies. The authors need to provide proof that these work as expected (WB with wt/mutant proteins, peptides, peptide-competitions experiments etc.).

We thank the reviewer for the suggestion. We performed Western blot and dot blot experiments to validate that the phospho-cGAS S120 antibody did work as expected.

Supplementary Fig. 3h

Minor points

1. Lines 141-145: The impact of the mutations interacting with DNA are potentially overinterpreted. As observed in PMID 31299200, a mutant more severe than the one used here only reduced affinity for DNA by two-fold. This should be rewritten.

We thank the reviewer for the suggestion. We have rewritten this sentence.

2. Fig. 1g, h, l, m: It is strange that loss of cGAS results in less production of the L1 reporter. This should be discussed.

We thank the reviewer for the comments. The experiments shown in Fig. 1l and m actually investigated the function of cGAS on endogenous L1 transcription (the cells were not transfected with the L1 reporter). As we replied to the major concern 2, we found that depletion of cGAS actually significantly decreased L1 mRNA levels, implying that there existed another layer of regulation on L1 transcription. One possible explanation is that there exists the negative feedback of ORF2p on L1 mRNA level. However, more experimental validation is required to draw a clear conclusion. We have discussed this point in our revised manuscript.

3. Fig. 1u, part of the label seems to be missing.

We thank the reviewer for the careful reading, and we have made the correction.

4. With regards to Fig. 2a, b, the authors should also test the flipside – protein stability, by treating cells with cycloheximide in order to shut down fresh Orf2 production.

We thank the reviewer for the suggestion. We examined the effect of cGAS depletion on ORF2p protein stability by treating cells with cycloheximide, and the data showed that knocking out cGAS significantly increased the protein stability of cGAS. The data have been included in our revised manuscript.

Supplementary Fig. 2a

5. The mutations that seem to impact interaction with CHK2 are all over cGAS when analysed on cGAS structures. This should be interpreted/discussed in the discussion.

We thank the reviewer for the professional comments. We demonstrated that three mutations on cGAS, S345L, L377P and P486L, impacted its interaction with CHK2. We agreed with the reviewer that the three residues dispersed on the primary structure of cGAS. However, when we mapped these three residues on the crystal structure (PDB 4O68) of cGAS, we found that S345, L377 and P486 are actually spatially proximal (as shown below), providing a plausible explanation for why their mutations could disrupt the physical association between the two proteins.

6. Extended Data Fig. 4d seems to not match the text.

We apologize for the confusion, and we have rephrased the sentence.

7. In Fig. 4c, HA-cGAS pulls down less Orf2 if it is L377P, but in Ext Fig. 4 ORF2 can pull down just as much L377P cGAS as wt. Can this be explained/discussed?

We apologize for any confusion caused. In Fig. 4c, whole cell lysate was extracted from cells transfected with wildtype or mutant cGAS, and used for Western blot analysis. The results showed that wildtype cGAS, rather than the L377P mutants, led to a decrease in ORF2p protein level. The blot of HA in Fig. 4c was displayed to confirm a comparable expression level between wildtype cGAS and its mutant. We have added descriptions on this point in the figure legends.

8. Please add MW labels to all WBs.

We thank the reviewer for the suggestion. We have added MW labels to all Western blots in our revised manuscript.

REVIEWER COMMENTS

Reviewer #1 (Remarks to the Author):

We appreciate the experiments conducted by the authors at the suggestion of both us and the other reviewers. We were also pleased to see many similar concerns raised by the other reviewers. Although the experiments performed do provide some resolve for certain concerns (namely the SILAC experiments), the author's inability to adequately respond to the concern shared by us and others that the contradictory data between L1 mRNA levels and ORF2p protein levels is problematic.

Rather than addressing these concerns, the authors acknowledge that their results contradict themselves and contend further understanding of cGAS regulation on L1 mRNA and ORF2 protein turnover is destined for further study. However, because the contention of the paper is that cGAS promotes ORF2p degradation and does simultaneously not regulate L1 mRNA transcription according to their qPCR data, the lack of data and explanation explaining this phenomenon greatly impedes the quality of this work. Ultimately, the interpretation of these results appears to be incorrect at some level and makes it impossible to fully believe the conclusions presented.

Additionally, as mentioned by us and others, this paper continues to lack substantial qualitative data and relies far too heavily on western blot analysis to draw rather dramatic conclusions. Many of the revisions and data added also happened to be western blots. This article requires more substantive data to draw the conclusions made by the authors.

Reviewer #2 (Remarks to the Author):

The authors have fully addressed my concerns. I support the publication of this manuscript.

Reviewer #3 (Remarks to the Author):

In their revised manuscript entitled, Zhen et al. have addressed many of my comments, and I particularly like the experiments that indicate increased abundance of endogenous L1s in the genome (Fig 1Sd-f). However, some issues, mostly relating to the Western Blots remain, and need to be addressed.

Major points

1. Western blots: It's great that the authors have the negative control stainings (tubulin/GAPDH) for the IPs. These should definitely be added to the figure. They are hugely important, because otherwise, the pulldowns cannot be interpreted.
2. Western blots: the authors should fix the patchwork western blots. Ideally, all samples from the same experiment should be on the same membrane. For the IP input and IP pulldowns, because all the loading controls at least seem to come from the same membranes I'd also be OK with the addition of the indication in the legends that inputs and pulldowns come from the same experiment, and if they come from different membranes. However, for Fig. 4e, f, all cGAS samples must be analysed on the same membrane.
3. The experiments with endogenous L1 elements (Fig. 1Sd-f) are particularly important, as almost all other experiments were done in artificial Orf2 overexpression or L1 reporter situations. I would suggest to put these in the main figures. I would also prefer the experiments from Fig. 4 to be done with this assay for endogenous L1.
4. The authors need to also determine the specificity of the commercial S305ph antibody. The company's documentation (<https://abclonal.co.uk/catalog-antibodies/PhosphoCGASS291RabbitpAb/AP1176>) does not show a proof that this antibody is phosphospecific, nor that it is specific to this particular site. Please also show the dot blots in the

supplementary figures, as they hold important information.

5. It is curious that if the mechanism the authors describe goes through Chk2, why it seems to act also under conditions of no DNA damage. This is rather unexpected, and indeed the authors show that under resting conditions, there is virtually no phospho-signal (for example Fig. S3i), yet the alanine mutants have a profound effect (for example Fig. 3n). This should at the very least be commented upon in the discussion and the Fig. 6 legends.

6. When describing Fig. S2d (main text and figure legends), the authors say that this is an experiment with purified proteins. However, in the figure, a GAPDH western blot is shown, which indicates that this is made from lysates? Therefore, the conclusion of a "direct" interaction is not warranted, and this should be modified in the text and in Fig. 6.

Minor points

1. The blue and green signal from the fluorescent western blots should be shown in greyscale. Shades of colour (and particularly blue) are much harder for the human eye to distinguish than shades of grey.

2. Fig. S3f: It seems like the mutant is missing the condition without etoposide.

REVIEWER COMMENTS

Reviewer #1 (Remarks to the Author):

We appreciate the experiments conducted by the authors at the suggestion of both us and the other reviewers. We were also pleased to see many similar concerns raised by the other reviewers. Although the experiments performed do provide some resolve for certain concerns (namely the SILAC experiments), the author's inability to adequately respond to the concern shared by us and others that the contradictory data between L1 mRNA levels and ORF2p protein levels is problematic.

Rather than addressing these concerns, the authors acknowledge that their results contradict themselves and contend further understanding of cGAS regulation on L1 mRNA and ORF2 protein turnover is destined for further study. However, because the contention of the paper is that cGAS promotes ORF2p degradation and does simultaneously not regulate L1 mRNA transcription according to their qPCR data, the lack of data and explanation explaining this phenomenon greatly impedes the quality of this work. Ultimately, the interpretation of these results appears to be incorrect at some level and makes it impossible to fully believe the conclusions presented.

We are deeply grateful for the reviewer's insightful attention to this matter. In our report, since we observed a cGAS-mediated repression of L1 retrotransposition, we first hypothesized that cGAS might exert this suppressive effect by suppressing L1 transcription, and therefore performed a series of experiments, including dual-luciferase assays and real-time qPCR to test this hypothesis. However, as we showed in the manuscript, our findings ruled out the possibility that cGAS repressed L1 retrotransposition by negatively regulating L1 transcription. Instead, as the reviewer noted, we observed a down-regulation in L1 transcription in cGAS KO cells. We find this observation intriguing, yet we feel that it does not undermine our discovery regarding the role of cGAS in promoting ORF2p degradation, and the regulations at transcriptional level and post-translational level are not contradictory.

However, we agree with the reviewer on this issue that the finding of cGAS potentially involved in regulating L1 transcription is of great interest. Therefore, as suggested by the reviewer, we carried out additional experiments to investigate the potential mechanisms responsible for the decreased L1 transcription following cGAS depletion. Firstly, a previous study conducted by Gentili et al. (*Cell Reports*, 2019) reported the enrichment of nuclear cGAS on LINE DNA repeats. Interestingly, subsequent analysis of the cGAS ChIP-seq data generated from this study indicated a preferential enrichment of cGAS on specific LINE subfamilies, including L1ME3A and L1MB3, as opposed to full-length L1. Further analysis revealed that cGAS does not bind to the 5' UTR region of L1, suggesting this transcriptional regulation of L1 by cGAS might not be a direct process.

[REDACTED]

Next, we analyzed publicly available RNA-seq data that had been previously published (Glück et al., *Nature Cell Biology*, 2017), and we observed that the expression of some transcription factors, known to regulate L1 transcription, altered upon cGAS depletion in mouse cells. These factors include *CTCF*, *RUNX3* (both are positive regulator of L1 transcription, reported by Sun et al., *PNAS*, 2018 and Yang et al., *Nucleic Acid Research*, 2003), and *DUSP1* (a negative regulator of L1 transcription, reported by Briggs et al, *Mobile DNA*, 2021).

[REDACTED]

Subsequently, we conducted RT-qPCR experiments to further investigate the expression levels of these transcription factors in *cGAS* knockout HeLa cells. Our data indicated a notable decrease in *CTCF* and *RUNX3* mRNA levels, consistently with the RNA-seq data, while *DUSP1* mRNA levels remained largely unchanged.

[REDACTED]

To deepen our understanding, we performed additional experiments analyzing L1 mRNA levels and promoter activity in *cGAS* knockout cells overexpressing either *CTCF* or *RUNX3*. Our findings revealed that overexpression of *CTCF* or *RUNX3* could rescue the decreased L1 promoter activity and L1 mRNA levels in *cGAS* knockout cells.

[REDACTED]

Moreover, we also investigated the impact of *DUSP1* knockdown on L1 mRNA levels and promoter activity in *cGAS* knockout cells. As expected, we found that neither L1 mRNA levels nor promoter activity were significantly influenced by the absence of *DUSP1*.

[REDACTED]

Therefore, we would like to propose that the deficiency of *CTCF* or *RUNX3* in *cGAS* knockout cells may lead to the observed decrease in L1 mRNA levels and promoter activity. Nevertheless, a number of questions still require to be investigated. For instances, through what mechanisms does *cGAS* regulate the transcription of these genes and in what biological contexts does this regulation occur? Moreover, is this potential regulation of physiological or pathological significance?

To clarify the narrative and avert any potential misunderstanding, we have thoughtfully decided to relocate this specific data which were displayed in Figure 1g, h, m, n in our previous version of manuscript, to Supplementary Figure 1, and only included the new results regarding the transcriptional regulation of L1 in the response letter. We hope this adjustment makes the presentation of our findings more coherent. However, if the reviewer or editor feels it necessary, we would add some of the data to our current manuscript.

Additionally, as mentioned by us and others, this paper continues to lack substantial qualitative data and relies far too heavily on western blot analysis to draw rather dramatic conclusions. Many of the revisions and data added also happened to be western blots. This article requires more substantive data to draw the conclusions made by the authors.

We sincerely appreciate the valuable feedback provided by the reviewer. In order to obtain more comprehensive qualitative data, we conducted proximity ligation assay (PLA) experiments. The results obtained from PLA experiments align with the Western blotting data presented in Figure 3l, further confirming that the two mutations in *cGAS* (S120A, S305A) indeed attenuate the *cGAS*-TRIM41 interaction.

Supplementary Fig. 4b

Furthermore, in line with the Western blotting results shown in Figure 3m, we observed that these two mutations abolished the *cGAS*-mediated stimulatory effect by interfering with the association of TRIM41-ORF2p.

Supplementary Fig. 4g

Consistent with the Western blotting results shown in Figure S5a-b and S5f-g, PLA assay demonstrated that the CHK2 inhibitor, BML-277 disrupted the enhanced association of cGAS with TRIM41, and TRIM41 with ORF2p, respectively, after DNA damage.

Supplementary Fig. 5c

Supplementary Fig. 5h

Additionally, our observations revealed that seven cancer-associated cGAS mutations were unable to facilitate the TRIM41-ORF2p interaction, as confirmed by the results obtained from PLA assay. These findings provide additional validation and support to our Western blotting results presented in Figure 5.

Supplementary Fig. 7i

We would like to express our sincere appreciation to the reviewer again for granting us the opportunity to enhance the strength of our conclusion.

Reviewer #2 (Remarks to the Author):

The authors have fully addressed my concerns. I support the publication of this manuscript.

We thank the reviewer a lot for the careful reading and positive feedback.

Reviewer #3 (Remarks to the Author):

In their revised manuscript entitled, Zhen et al. have addressed many of my comments, and I particularly like the experiments that indicate increased abundance of endogenous L1s in the genome (Fig 1Sd-f). However, some issues, mostly relating to the Western Blots remain, and need to be addressed.

We thank the reviewer for the positive comments.

Major points

1. Western blots: It's great that the authors have the negative control stainings (tubulin/GAPDH) for the IPs. These should definitely be added to the figure. They are hugely important, because otherwise, the pulldowns cannot be interpreted.

We thank the reviewer for the suggestion. We have included the negative control staining of tubulin or GAPDH for all IPs in our revised manuscript.

2. Western blots: the authors should fix the patchwork western blots. Ideally, all samples from the same experiment should be on the same membrane. For the IP input and IP pulldowns, because all the loading controls at least seem to come from the same membranes I'd also be OK with the addition of the indication in the legends that inputs and pulldowns come from the same experiment, and if they come from different

membranes. However, for Fig. 4e, f, all cGAS samples must be analysed on the same membrane.

We thank the reviewer for the suggestion. In our revised manuscript, we have included a statement in the figure legends to clarify that all the inputs and pulldowns came from the same experiments, including the experiments shown in Fig. 4e-f.

Regarding Figure 4e-f, we have taken into consideration the significant difference in cGAS protein levels between the cytoplasm and nucleus, thus the cropped blots were shown in our previous version of manuscript. To clearly demonstrate the alterations of cGAS protein in control and SIPS cells, we have provided images with different exposure times in our revised manuscript.

3. The experiments with endogenous L1 elements (Fig. 1Sd-f) are particularly important, as almost all other experiments were done in artificial Orf2 overexpression or L1 reporter situations. I would suggest to put these in the main figures. I would also prefer the experiments from Fig. 4 to be done with this assay for endogenous L1.

We thank the reviewer a lot for the suggestion. We have moved the experiments related to endogenous L1 copy number to Figure 1 in our revised manuscript. Furthermore, we conducted qPCR experiments to assess L1 copy numbers in senescent cells. Our results revealed that control cells exhibited an increased genomic L1 DNA content compared to senescent wildtype cells. Moreover, the depletion of cGAS led to a significant increase in genomic L1 DNA content in senescent cells. These additional experiments provide further evidence supporting the crucial role of cGAS in inhibiting L1 retrotransposition in senescent cells.

Supplementary Fig. 6f

Supplementary Fig. 6g

4. The authors need to also determine the specificity of the commercial S305ph antibody. The company's documentation (<https://abclonal.co.uk/catalog-antibodies/PhosphoCGASS291RabbitAb/AP1176>) does not show a proof that this antibody is phosphospecific, nor that it is specific to this particular site. Please also show the dot blots in the supplementary figures, as they hold important information.

We thank the reviewer for the suggestion. The commercial S305ph antibody was customized by (Zhong et al., *Cell Discovery*, 2020), and the specificity of this antibody was validated through dot blot experiments in the referenced paper. In addition, as suggested by the reviewer, we also performed Western blot experiments to further confirm the functionality and reliability of this antibody.

Supplementary Fig. 3i

5. It is curious that if the mechanism the authors describe goes through Chk2, why it seems to act also under conditions of no DNA damage. This is rather unexpected, and indeed the authors show that under resting conditions, there is virtually no phospho-signal (for example Fig. S3i), yet the alanine mutants have a profound effect (for example Fig. 3n). This should at the very least be commented upon in the discussion and the Fig. 6 legends.

We appreciate the reviewer's insightful point. We hypothesized that cells may experience endogenous DNA damage resulting from proliferation or reactive oxygen species (ROS) generated during metabolism, as it has been reported that approximately 25 breaks are generated per human cell per day (Tubbs et al., *Cell*, 2019). Actually, even

under normal conditions, cells exhibited a basal level of cGAS phosphorylation at both S120 and S305 residues. To clearly visualize the phosphorylation signal under resting conditions, we have replaced the original image with a new one that was exposed for a longer duration in Fig. S3i (currently Fig. 3j). We have included the discussion regarding this issue in our revised manuscript.

6. When describing Fig. S2d (main text and figure legends), the authors say that this is an experiment with purified proteins. However, in the figure, a GAPDH western blot is shown, which indicates that this is made from lysates? Therefore, the conclusion of a “direct” interaction is not warranted, and this should be modified in the text and in Fig.

We apologize for any confusion caused. It is now clear that the experiments presented in Figure S2d were performed *in vitro*. We have rectified this error. Thank you for bringing this to our attention, and we apologize for this issue again.

Minor points

1. The blue and green signal from the fluorescent western blots should be shown in greyscale. Shades of colour (and particularly blue) are much harder for the human eye to distinguish than shades of grey.

We thank the reviewer for the suggestion. We have changed the fluorescent western blots into greyscale.

2. Fig. S3f: It seems like the mutant is missing the condition without etoposide.

We thank the reviewer for the suggestion. Actually, we found that D347A mutation also reduced the association between cGAS and CHK2 under the normal conditions. The data have been included in our revised manuscript.

Fig. 3g

Supplementary Fig. 3f

REVIEWER COMMENTS

Reviewer #1 (Remarks to the Author):

As before, we sincerely appreciate the experiments performed by the authors to address the issues raised by both us and other reviewers about the findings of this manuscript. The interactions between cGAS, LINE1 mRNA transcripts and its associated proteins, and inflammation is of tremendous interest and importance. The author's finding that cGAS promotes and facilitates L1 ORF2p degradation via TRIM41 is an exciting, very important discovery.

We also very much appreciate the experiments performed attempting to answer the question regarding the seemingly contradictory results displaying increased ORF2p expression in cGAS KO cells but decreased L1 mRNA. We find the idea of differing transcription factors expression causing the observed phenotype is an interesting approach, but it is difficult to fully accept the findings the authors hope to claim (that the higher ORF2p levels in cGAS KO cells is occurring irrelevant of lower L1 mRNA transcription) based on just qPCR and publicly available RNAseq data.

Because of how difficult it appears to fully address the concern of differing ORF2p levels and L1 mRNA levels, we would like to simply propose the authors move forward with publishing the article demonstrating ORF2p degradation via TRIM41 is enhanced by nuclear cGAS. However, we believe it would be misleading to suggest that cGAS depletion reduces L1 mRNA levels as the presented data is not definitive enough on this matter.

Therefore, we propose the removal of the sentence starting on line 170 to 171 stating cGAS depletion decreased L1 mRNA levels given how weak the results that lead to that conclusion are. Instead, we suggest the authors focus primarily on the cGAS and ORF2p/TRIM41 interaction and acknowledge that the exact mechanisms involved are to be uncovered at another time without excluding the possibility of L1 mRNA expression changes during cGAS depletion.

We appreciate the willingness of the authors to add additional experiments to diversify their findings and are satisfied regarding that concern.

Reviewer #3 (Remarks to the Author):

In their revised manuscript entitled, "Nuclear cGAS restricts L1 retrotransposition by promoting TRIM41-mediated ORF2p ubiquitination and degradation", Zhen et al. have addressed almost all issues that I raised, but prior to publication a few minor issues with the co-IP experiments need to be fixed.

Minor points:

1. For many of the co-IP experiments, the authors still do not show negative controls (Fig. 3b, 2i, S2j, S2m, S4e, S4f, S5g, S5k, S7d, e, g, h). Please add these.
2. With regards to Fig. S2d (lines 188-190), the authors again talk about "purified" proteins and "direct" interactions. But this is still not unambiguously shown. While cGAS has indeed been prepared with purification methods, Orf2p was generated by a single-step pulldown from overexpression in HEK293T cells. This preparation procedure is not extensive enough to warrant the term "purified", and therefore the conclusion of a "direct" interaction is not warranted at this point, as components co-purified with Orf2p could bridge the interaction with cGAS. The same goes for the interaction between Orf2p and Trim41 (lines 208/209, Fig. 2i). Please modify the text.
3. The connection with DNA damage I pointed out in my previous review should be discussed. At the moment it hasn't been explained why CHK2 is involved under conditions without DNA damage. Similarly, the Y215 mutation experiment requires additional discussion. In their previous work, the authors suggested that Y215ph-dependent regulation was exclusive to DNA damage conditions, so one wouldn't expect the Y215E mutation to have an effect on L1 under conditions without DNA damage, but this is what Fig. 3 shows. Perhaps the mutant only affects a subset of cells in which endogenous damage rises to levels above the threshold required to activate CHK2?

4. Lines 49-51: " Recently, cumulating evidence showed that cGAS was also present in the nucleus, while the non-canonical positive function of nuclear cGAS has not yet been fully understood". The meaning of this sentence is not clear to me.
5. Reading the text relating to Fig. S1g-i is confusing, as one might have expected loss of cGAS to increase transcript levels. I think it's fine to leave this issue unresolved in this paper, but the authors need to point out that this is not what one might have expected.
6. Lines 261. 296: The word "proved" should be exchanged with "indicated" and "provide evidence", respectively. "Prove" implies that alternative explanations are not possible, but in biology this level of certainty is very hard to achieve.
7. The inverted greyscale of the fluorescent western blots is confusing and doesn't add any useful information. Please convert to regular greyscale.
8. Lines 331-333: The use of the word "significant" is misleading as it can mean both "statistically significant" as well as "substantial". While the result seems to indeed show an effect of "statistical significance" it cannot be described as "substantial". Please change to something like "...we observed a small, but statistically significant, increase...".

REVIEWER COMMENTS

Reviewer #1 (Remarks to the Author):

As before, we sincerely appreciate the experiments performed by the authors to address the issues raised by both us and other reviewers about the findings of this manuscript. The interactions between cGAS, LINE1 mRNA transcripts and its associated proteins, and inflammation is of tremendous interest and importance. The author's finding that cGAS promotes and facilitates L1 ORF2p degradation via TRIM41 is an exciting, very important discovery.

We also very much appreciate the experiments performed attempting to answer the question regarding the seemingly contradictory results displaying increased ORF2p expression in cGAS KO cells but decreased L1 mRNA. We find the idea of differing transcription factors expression causing the observed phenotype is an interesting approach, but it is difficult to fully accept the findings the authors hope to claim (that the higher ORF2p levels in cGAS KO cells is occurring irrelevant of lower L1 mRNA transcription) based on just qPCR and publicly available RNAseq data.

Because of how difficult it appears to fully address the concern of differing ORF2p levels and L1 mRNA levels, we would like to simply propose the authors move forward with publishing the article demonstrating ORF2p degradation via TRIM41 is enhanced by nuclear cGAS. However, we believe it would be misleading to suggest that cGAS depletion reduces L1 mRNA levels as the presented data is not definitive enough on this matter.

Therefore, we propose the removal of the sentence starting on line 170 to 171 stating cGAS depletion decreased L1 mRNA levels given how weak the results that lead to that conclusion are. Instead, we suggest the authors focus primarily on the cGAS and ORF2p/TRIM41 interaction and acknowledge that the exact mechanisms involved are to be uncovered at another time without excluding the possibility of L1 mRNA expression changes during cGAS depletion.

We appreciate the willingness of the authors to add additional experiments to diversify their findings and are satisfied regarding that concern.

We sincerely thank the reviewer for the patience and valuable assistance throughout the revision process. As the reviewer suggested, we removed the sentence from line 170 to 171. Instead, we have clarified in the revised manuscript that our primary focus is to elucidate the role of cGAS in regulating L1 at the protein level. However, the impact of cGAS on L1 transcription remains uncertain and necessitates further investigations. This clarification is now included in lines 187 to 189 of our revised manuscript.

Reviewer #3 (Remarks to the Author):

In their revised manuscript entitled, “Nuclear cGAS restricts L1 retrotransposition by promoting TRIM41-mediated ORF2p ubiquitination and degradation”, Zhen et al. have addressed almost all issues that I raised, but prior to publication a few minor issues with the co-IP experiments need to be fixed.

We thank the reviewer for the positive comments.

Minor points:

1. For many of the co-IP experiments, the authors still do not show negative controls (Fig. 3b, 2i, S2j, S2m, S4e, S4f, S5g, S5k, S7d, e, g, h). Please add these.

We thank the reviewer for pointing out the issues. In our revised manuscript, we have added negative controls for all of these co-IP experiments, except for 2i, as it was performed *in vitro*. These controls are displayed in the respective panels and are also included in the source data files.

2. With regards to Fig. S2d (lines 188-190), the authors again talk about “purified” proteins and “direct” interactions. But this is still not unambiguously shown. While cGAS has indeed been prepared with purification methods, Orf2p was generated by a single-step pulldown from overexpression in HEK293T cells. This preparation procedure is not extensive enough to warrant the term “purified”, and therefore the conclusion of a “direct” interaction is not warranted at this point, as components co-purified with Orf2p could bridge the interaction with cGAS. The same goes for the interaction between Orf2p and Trim41 (lines 208/209, Fig. 2i). Please modify the text.

We thank the reviewer for the comment. We have rephrased the text as the reviewer suggested.

3. The connection with DNA damage I pointed out in my previous review should be discussed. At the moment it hasn't been explained why CHK2 is involved under conditions without DNA damage. Similarly, the Y215 mutation experiment requires additional discussion. In their previous work, the authors suggested that Y215ph-dependent regulation was exclusive to DNA damage conditions, so one wouldn't expect the Y215E mutation to have an effect on L1 under conditions without DNA damage, but this is what Fig. 3 shows. Perhaps the mutant only affects a subset of cells in which endogenous damage rises to levels above the threshold required to activate CHK2?

We thank the reviewer for the comments. We have added more discussion on this point. The relative description is now included in line 311 to 315 of our revised manuscript.

4. Lines 49-51: ” Recently, cumulating evidence showed that cGAS was also present in the nucleus, while the non-canonical positive function of nuclear cGAS has not yet been fully understood”. The meaning of this sentence is not clear to me.

We thank the reviewer for the comment. We have revised the text as follows: “Recently, cumulating evidence revealed the presence of cGAS within the nucleus. However, the biological functions of nuclear cGAS have not been fully understood.” in our revised manuscript.

5. Reading the text relating to Fig. S1g-i is confusing, as one might have expected loss of cGAS to increase transcript levels. I think it’s fine to leave this issue unresolved in this paper, but the authors need to point out that this is not what one might have expected.

We thank the reviewer for the valuable suggestion. Taking into account the advice from both Reviewer 1 and Reviewer 3, to avoid any confusion, we decided to remove the data regarding the role of cGAS on L1 transcription from our revised manuscript. We have expanded the discussion on the importance and need for further investigation in this area. This clarification has been incorporated into lines 187 to 189 of our revised manuscript.

6. Lines 261. 296: The word “proved” should be exchanged with “indicated” and “provide evidence”, respectively. “Prove” implies that alternative explanations are not possible, but in biology this level of certainty is very hard to achieve.

We thank the reviewer for the careful reading. We have edited the texts as the reviewer suggested.

7. The inverted greyscale of the fluorescent western blots is confusing and doesn’t add any useful information. Please convert to regular greyscale.

We thank the reviewer for the valuable suggestion. We have converted these panels to regular greyscale in our revised manuscript.

8. Lines 331-333: The use of the word “significant” is misleading as it can mean both “statistically significant” as well as “substantial”. While the result seems to indeed show an effect of “statistical significance” it cannot be described as “substantial”. Please change to something like “...we observed a small, but statistically significant, increase...”.

We thank the reviewer for the careful reading. We have edited the texts as the reviewer suggested.